# Pharmacogenomics polygenic risk score for drug response prediction using PRS-PGx methods

Song Zhai[1,3], Hong Zhang [1,3], Devan V. Mehrotra[2] & Judong Shen [1]✉

Polygenic risk scores (PRS) have been successfully developed for the prediction of human diseases and complex traits in the past years. For drug response prediction in randomized clinical trials, a common practice is to apply PRS built from a disease genome-wide association study (GWAS) directly to a corresponding pharmacogenomics (PGx) setting. Here, we show that such an approach relies on stringent assumptions about the prognostic and predictive effects of the selected genetic variants. We propose a shift from disease PRS to PGx PRS approaches by simultaneously modeling both the prognostic and predictive effects and further make this shift possible by developing a series of PRS-PGx methods, including a novel Bayesian regression approach (PRS-PGx-Bayes). Simulation studies show that PRS-PGx methods generally outperform the disease PRS methods and PRS-PGx-Bayes is superior to all other PRS-PGx methods. We further apply the PRS-PGx methods to PGx GWAS data from a large cardiovascular randomized clinical trial (IMPROVE-IT) to predict treatment related LDL cholesterol reduction. The results demonstrate substantial improvement of PRS-PGx-Bayes in both prediction accuracy and the capability of capturing the treatment-specific predictive effects while compared with the disease PRS approaches.

Pharmacogenomics (PGx), an important tool for precision medicine, studies how pharmacokinetics, pharmacodynamics, efficacy, and safety responses to drugs are associated with genetic information at the molecular level of treated subjects[1–3]. Efficacy PGx studies have great potential to guide treatment options by integrating routine pharmacogenomic screening into clinical development and proposing novel strategies for identifying genetic markers that impact efficacy for new compounds (and for marketed drugs, if applicable)[4]. In the domain of precision medicine, many associations between genetic variation and inter-individual difference in drug response have been discovered to tailor treatments to the genetic makeup of the patient[4]. However, the conventional single variant PGx biomarkers or drug response predictors usually rely on large effect sizes. Genetic variants with small but genuine effects may not reach the significance threshold in a typical PGx analysis of hundreds to a

few thousand subjects. Recent developments in disease genetics reveal that the polygenicity, i.e., many small genetic effects, is present in many complex traits[5]. This observation provides evidence to support modeling and predicting disease status by combining the effects of many weak signals.

Polygenic risk score (PRS), defined as the weighted sum of the effect sizes of many polymorphisms, is a rapidly emerging tool in the disease genetics field. PRS reflects the overall genetic risk of a phenotype of interest and can be used as a stratification mechanism for downstream analyses and decision making. In disease genetics, PRSs have been successfully developed for multiple complex diseases including coronary artery disease[6,7], cancer[8,9], etc. A large variety of methods have been developed for constructing PRS. To name a few, they include (1) the unadjusted method which builds PRS using the unadjusted effect estimates of SNPs across whole genome; (2) the

[1]Biostatistics and Research Decision Sciences, Merck & Co., Inc., Rahway, NJ 07065, USA. [2]Biostatistics and Research Decision Sciences, Merck & Co., Inc., North Wales, PA 19454, USA. [3]These authors contributed equally: Song Zhai, Hong Zhang. ✉e-mail: judong.shen@merck.com

Clumping and Thresholding (C+T) method[10] which builds a few PRSs using independent SNPs that pass different significance thresholds and the optimal threshold is selected according to their performance in an independent study; (3) the Lassosum method[11] which uses penalized regression to select informative SNPs by incorporating linkage disequilibrium (LD) information, and (4) the Bayesian regression methods, e.g., LDpred[12], LDpred2[13], and PRS-CS[14], which shrink the marginal effect sizes with respect to LD. Among them, the more sophisticated penalized regression and Bayesian regression methods have been shown to achieve better performance over the C+T method[13].

Like many complex traits, most drug responses in PGx are extremely polygenic[3,15]. Despite recent development in PRS methods and their exciting applications in disease genetics, similar analytic methods have largely not yet been successfully adapted to drug responses in PGx[16]. There are emerging examples published so far, which build PRS from disease GWAS using SNPs with treatment unrelated prognostic effects only and then test whether the PRS is predictive of drug responses in one or several PGx studies[17–21]. However, this current practice of building disease PRS and applying to PGx data (called PRS-Dis approach) has not been fully justified in theory. In fact, we can show (Results section) that the PRS-Dis approach relies on a very stringent assumption that every variant selected for constructing PRS should have a constant ratio between its genotype main effect and genotype-by-treatment interaction effect, which may not be true in real PGx data. On the other hand, to the best of our knowledge, only a few published studies directly build PRS from drug-related data for safety or efficacy PGx prediction. For example, Lanfear et al.[16] build an efficacy PGx PRS for $\beta$-blockers using observational data. Koido et al.[22] build a PGx PRS for drug-induced liver injury using a method similar to C+T. Lewis et al.[23] build an efficacy PGx PRS for clopidogrel response in terms of cardiovascular outcomes using single-arm clinical data. However, there is limited methodological development on the adaptation of PRS methods in disease genetics to PGx where data from both treatment and placebo (or control) arms are available.

To tackle the challenges of the complex drug response prediction and the lack of state-of-the-art PGx PRS methods, we propose to shift from the disease PRS approach to the PGx PRS approach by jointly modeling the genetic main effect and the genotype-by-treatment interaction effect (called PRS-PGx approach). We systematically extend the current PRS-Dis methods to construct both prognostic and predictive PRSs for drug response prediction in PGx studies. These methods include PRS-PGx-Unadj (Unadjusted), PRS-PGx-CT (Clumping + Thresholding), PRS-PGx-L, -GL, -SGL (-Lasso, -Group Lasso, -Sparse Group Lasso), and PRS-PGx-Bayes (Bayesian regression) methods. Our proposed methods use only PGx genome-wide association summary statistics and an external LD reference panel except for the penalized regression-based methods, which require access to individual level genetic and phenotypic data. Moreover, by extending the idea of global-local scaling parameters from disease GWAS[14] to PGx GWAS, the PRS-PGx-Bayes method is able to infer the posterior prognostic and predictive effects simultaneously.

Our simulation studies demonstrate that PRS-PGx methods generally outperform the PRS-Dis methods across a wide range of genetic architectures and PRS-PGx-Bayes is superior to all other PRS-PGx methods. These methods are further applied to the IMPROVE-IT (IMProved Reduction of Outcomes: Vytrocin Efficacy International Trial)[24] PGx GWAS summary statistics data[25] to predict treatment-related LDL cholesterol reduction. The drug response prediction results demonstrate a substantial improvement of PRS-PGx-Bayes in both prediction accuracy and the capability of capturing the predictive effect over alternative methods.

## Results

### Conceptual framework of the PRS-PGx methods

We consider a high-dimensional regression model of $n$ patients and $m$ SNPs for a drug response:

$$\mathbf{Y} = \mathbf{X}\gamma + \beta_T \mathbf{T} + \mathbf{G}\beta + (\mathbf{G} \times \mathbf{T})\alpha + \epsilon, \quad (1)$$

where $\mathbf{Y}$ denotes a quantitative trait (drug response), $\mathbf{T}$ the binary treatment assignment, $\mathbf{X}$ the n × p matrix of covariates, and $\mathbf{G}$ the n × m genotype matrix; $\beta$ is a m × 1 vector of prognostic effects (i.e., main effects), $\alpha$ is a m × 1 vector of predictive effects (i.e., interaction effects), and $\epsilon$ is the random error. In practice, the phenotype $\mathbf{Y}$ can first be adjusted by the covariates $\mathbf{X}$ and the treatment $\mathbf{T}$, before application to any PRS-PGx algorithms. For simplicity, we will use $\mathbf{Y}$ as the phenotype after such adjustment in the later discussion.

The regression coefficient $\mathbf{b} = (\beta, \alpha)$ is assumed to be fixed in the PRS-PGx-Unadj, PRS-PGx-CT and PRS-PGx-L, -GL, -SGL methods and random in the PRS-PGx-Bayes method. Specifically, for each $j = 1, \cdots, m$, we consider the following prior distribution of $\mathbf{b}_j = (\beta_j, \alpha_j)$ in the Bayesian approach:

$$\begin{pmatrix} \beta_j \\ \alpha_j \end{pmatrix} | \sigma^2, \phi, \psi_j, \xi_j, \rho_j \sim MVN\left(\mathbf{0}, \frac{\sigma^2}{n}\phi M_j\right), \text{where } M_j = \begin{bmatrix} \psi_j & \rho_j\sqrt{\psi_j \xi_j} \\ \rho_j\sqrt{\psi_j \xi_j} & \xi_j \end{bmatrix} \sim g, \quad (2)$$

where $\phi$ is a global scaling parameter that is shared across multiple SNPs and controls the degree of the model sparseness; $\psi_j$ and $\xi_j$ are local and marker-specific scaling parameters; $\rho_j$ is the marker-specific correlation between the two effect sizes $\beta_j$ and $\alpha_j$; and $g$ is a probability density function of a random matrix.

In PRS-PGx methods, SNPs are used for the construction of prognostic PRS and predictive PRS based on their estimated $\mathbf{b}^{est} = (\beta^{est}, \alpha^{est})$. The prognostic and predictive PRSs are defined as the weighted sum of the selected SNPs' genotypes, where the weights are the estimated prognostic and predictive effect sizes, respectively,

$$S_{prog} = \sum_{j=1}^{m} \beta_j^{est} \mathbf{G}_j, \quad S_{pred} = \sum_{j=1}^{m} \alpha_j^{est} \mathbf{G}_j. \quad (3)$$

The predictive PRS is useful for patient stratification by aggregating the differential treatment effects. We can define the PGx PRS as

$$S_{PGx} = \begin{cases} S_{prog} + S_{pred}, & T = 1, \\ S_{prog}, & T = 0, \end{cases} \quad (4)$$

for overall drug response prediction. More technical details are provided in the "Methods" section.

### Assumption of PRS-Dis approach for drug response prediction

Consider the linear model defined in Eq. (1). Assume (i) SNPs $\mathbf{G}_i, i = 1, \ldots, m$ are standardized: $E\mathbf{G}_i = 0$, $var(\mathbf{G}_i) = 1$; (ii) $E(\epsilon) = 0$, $var(\epsilon) = \sigma^2$; (iii) $\mathbf{b}$ is defined in Eq. (2); (iv) $\mathbf{G}_i, i = 1, \ldots, m$, $\mathbf{b}$, and $\epsilon$ are mutually independent. We consider the following three quantities:

$$Y|(T = 1) = \sum_{i=1}^{m} (\beta_i + \alpha_i)\mathbf{G}_i + \epsilon, \quad (5)$$

$$S_{PGx}|(T = 1) = \sum_{i=1}^{m} (\beta_i + \alpha_i)\mathbf{G}_i, \quad (6)$$

$$S_{Dis} = \sum_{i=1}^{m} \beta_i \mathbf{G}_i. \quad (7)$$

$Y|(T = 1)$ is the observed response of a subject in the treatment arm with SNPs $\mathbf{G}_i$, $i = 1, \ldots, m$; $S_{\text{Dis}}$ and $S_{\text{PGx}}|(T = 1)$ are the *perfect* polygenic scores for this treated subject from disease GWAS and PGx, respectively. We will drop the condition notation '$|(T = 1)$' hereafter when there is no ambiguity. We prove (in Supplementary Method A) that the heritability of a drug response can be calculated as

$$h^2 = \frac{\text{var}\left(\sum_{i=1}^{m}(\beta_i + \alpha_i)\mathbf{G}_i\right)}{\text{var}\left(\sum_{i=1}^{m}(\beta_i + \alpha_i)\mathbf{G}_i\right) + \sigma^2} = \text{cor}^2(S_{\text{PGx}}, Y). \quad (8)$$

On the other hand, it can be shown (in Supplementary Method B) that the squared correlation coefficient between $S_{\text{Dis}}$ and $Y$ for the treated subjects is:

$$\text{cor}^2(S_{\text{Dis}}, Y) = h^2 \left(1 - \frac{\sum_{i=1}^{m}\psi_i \sum_{i=1}^{m}\xi_i - \left(\sum_{i=1}^{m}\rho_i\sqrt{\psi_i\xi_i}\right)^2}{\left(\sum_{i=1}^{m}\psi_i\right)^2 + 2\sum_{i=1}^{m}\psi_i\sum_{i=1}^{m}\rho_i\sqrt{\psi_i\xi_i} + \sum_{i=1}^{m}\psi_i\sum_{i=1}^{m}\xi_i}\right). \quad (9)$$

In the scenario that all interaction effects are independent of main effects, i.e., $\rho_i \equiv 0$, $i = 1, \ldots, m$, Eq. (9) is reduced to $h^2\left(1 - \frac{1}{1+R}\right)$, where $R = \sum_{i=1}^{m}\psi_i/\sum_{i=1}^{m}\xi_i$ is the ratio between the total main effect and the total interaction effect. If $R \to \infty$, that is no interaction effect at all, then $\text{cor}^2(S_{\text{Dis}}, Y) \to h^2$. If, however, we have strong interaction effect and no main effect, $R \to 0$, then $\text{cor}^2(S_{\text{Dis}}, Y) \to 0$. This observation is consistent with the intuition that disease PRS approach, which ignores the treatment-by-genotype effects, is less ideal if such effects are strongly present.

In fact, by Cauchy-Schwarz inequality, $\text{cor}^2(S_{\text{Dis}}, Y) \le h^2$ and the equality holds if and only if

$$\rho_i \equiv 1 \text{ and } \psi_i \propto \xi_i, \quad \text{for all } i = 1, \cdots, m, \quad (10)$$

which is equivalent to

$$\beta_i = c\alpha_i, \quad i = 1, \cdots, m, \text{ for some constant number } c. \quad (11)$$

This explicitly shows that the disease PRS approach $S_{\text{Dis}}$ works only under an extremely stringent assumption that every causal variant must have the same interaction effect proportionate to its main effect. We also consider the situations when the regression coefficients $\beta_i$, $\alpha_i$, $i = 1, \ldots, m$, are fixed constants (in Supplementary Method B). The proof also shows that disease PRS $S_{\text{Dis}}$ cannot recover all heritability as long as the interaction effect is not proportionate to its main effect for all causal variants.

By using the IMPROVE-IT PGx GWAS summary statistics data and 1000 Genomes (1KG) Phase 3 data (http://csg.sph.umich.edu/abecasis/mach/download/1000G.Phase3.v5.html) as external reference panel, we can calculate the $\text{cor}^2(S_{\text{Dis}}, Y) = h^2(1 - 0.54)$, which means the PRS developed from any disease GWAS can at most explain 46% genetic variability of the drug response. In addition, we also calculate the ratio of genetic main effect to interaction effect, $c$, for the SNPs (after clumping with 250 kb window size and LD $r^2 > 0.8$) across whole genome ($m = 8,551,930$) and the top SNPs defined by $p$-values of 2df (joint G and G × T) two-sided test[26] less than three thresholds 1e−06 ($m = 16$), 1e−05 ($m = 81$), and 1e−04 ($m = 472$), respectively. Figure 1 shows that the constant ratio assumption (11) is completely not satisfied. Therefore, it is expected that the performance of PRS-Dis methods will be lower when applied to analyzing real PGx data (i.e., the IMPROVE-IT PGx GWAS data). Our real data analysis results indeed show that the PRS-Dis methods have substantially lower predictive power than the PRS-PGx methods.

## Simulation studies

In this section, we further illustrate the limitations of PRS-Dis methods and compare their empirical performance with the proposed PRS-PGx

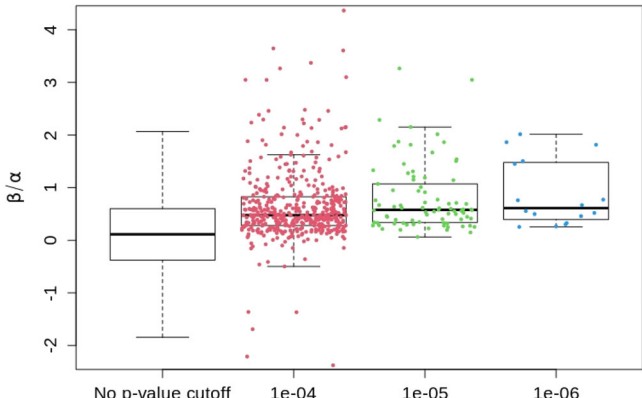

**Fig. 1 | Distributions of the prognostic to predictive effect size ratios calculated from the IMPROVE-IT PGx GWAS summary statistics data with $n = 5661$ unrelated European samples.** The left boxplot shows the distribution of whole genome SNPs (after clumping, $m = 8,551,930$). The right three boxplots show the distribution of top SNPs (after clumping) with their 2df (G + G × T) two-sided test $p$-values less than the three $p$-value thresholds 1e−06 ($m = 16$), 1e−05 ($m = 81$), and 1e−04 ($m = 472$), respectively. In each boxplot, the band indicates the median, the box indicates the first and third quartiles, and the whiskers indicate ± 1.5 × interquartile range. Effect size ratios of $m$ SNPs are overlaid on the corresponding boxplot as dot points.

methods. We considered the scenario where all causal variants were both prognostic and predictive and their effect sizes were positively correlated. The constant ratio assumption (11) was only partially satisfied since the correlation coefficients were assumed to follow uniform distribution. As a sensitivity analysis, we also considered the scenario where all causal variants were either prognostic or predictive but could not be both, i.e., the constant ratio assumption (11) was strongly violated.

**The constant ratio assumption of PRS-Dis is partially satisfied.** We simulated SNPs' prognostic and predictive effects from a bivariate normal distribution, as described in Eq. (15). Specifically, we set the heritability $H^2 = 0.3$, the treatment effect $\beta_T = 0$, and the prognostic and predictive effect sizes at the same scale with $\psi/\xi = 1$. Note that although $\psi_i \propto \xi_i$ for all $i \in \mathcal{I}$, where $\mathcal{I}$ is the set of causal variants, the correlation coefficient between the two effects $\rho_i$ ~ Uniform(0,1) may vary from different LD blocks. The full details of data generation process are provided in the "Methods" section.

Before we compared PRS-PGx and PRS-Dis methods, we first assessed the performance among the three disease PRS methods (PRS-Dis-Unadj, PRS-Dis-CT, and PRS-Dis-LDpred2) and the three machine learning-based PRS-PGx methods (PRS-PGx-L, PRS-PGx-GL, and PRS-PGx-SGL), respectively. As shown in Supplementary Fig. 1, among the three disease PRS methods, PRS-Dis-LDpred2 outperformed the others in terms of both $R^2$ and the statistical significance of its predictive effect. Similarly, among three penalized regression approaches, PRS-PGx-GL was consistently favored in the current simulation setting. Therefore, in the remaining simulations and real data analyses, we focused on only PRS-Dis-LDpred2 among all PRS-Dis methods; and only PRS-PGx-GL among all the PGx penalized regression methods.

Five polygenic prediction methods, PRS-Dis-LDpred2, PRS-PGx-Unadj, PRS-PGx-CT, PRS-PGx-GL, and PRS-PGx-Bayes, were compared across different settings of sample sizes, number of causal variants, heritabilities and effect sizes. The tuning parameters such as the p-value threshold in PRS-PGx-CT, the penalty parameter in PRS-PGx-GL, and some prior distribution parameters in PRS-PGx-Bayes were selected via 5-fold cross-validation (CV). The 1000 Genomes Project European population data was used as an external reference panel for LD. The performance was evaluated in an independent testing set

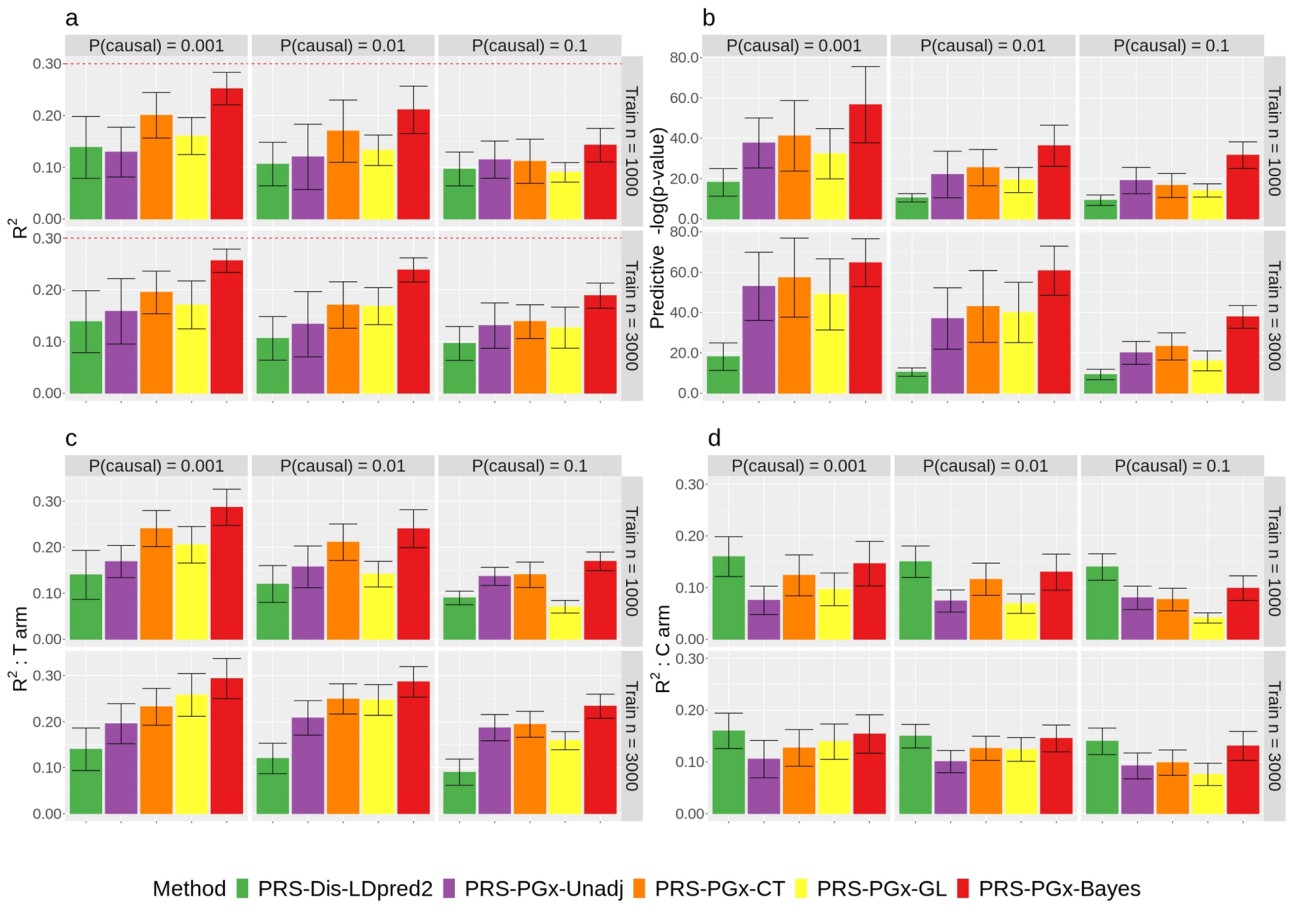

**Fig. 2 | Predictive performance of five polygenic prediction methods in the simulation studies, where heritability was fixed at 0.3 and $\psi/\xi = 1$.** The numbers of the causal variants for P(causal) = 0.001, 0.01, and 0.1 were 5, 50, and 500, respectively. The training sample size for PRS-PGx approaches was either 1000 or 3000; for PRS-Dis-LDpred2 approach was 20,000. The tuning parameters were selected via cross-validation in the training data. The performance was assessed in terms of **a** prediction accuracy $R^2$ of $S_{PGx}$ in two arms, **b** predictive $p$-value for the two-sided $S_{pred} \times T$ interaction test, **c** $R^2$ of $S_{PGx}$ under treatment arm, and **d** $R^2$ of $S_{PGx}$ under control arm. Data are presented as mean values +/− standard deviations (error bars) with 10,000 replications, where results were calculated from the testing sets.

(sample size = 1000) in terms of (i) the prediction accuracy of $S_{PGx}$ quantified by $R^2$ between the observed and predicted phenotypes in two arms; (ii) the predictive effect measured by the $-\log_{10}(p\text{-value})$ from the two-sided likelihood ratio test (LRT) of $S_{pred} \times T$ interaction; and iii) $R^2$ of the $S_{PGx}$ under treatment and control arms, respectively. The statistical details about the above analyses are provided in the "Methods" section.

The predictive performance of the five polygenic prediction methods from the simulation studies is summarized in Fig. 2a. The PRS-PGx methods generally outperformed the PRS-Dis method (i.e., PRS-Dis-LDpred2). Among PRS-PGx methods, our proposed Bayesian approach PRS-PGx-Bayes was consistently better than the others. Overall speaking, PRS-PGx-Unadj approach, which aggregated all SNPs, performed poorly when the number of causal variants was small, but became more comparable to other methods when the genetic architectures were highly polygenic. Although it is reasonable to expect that PRS-PGx-GL (which accounts for local LD patterns) likely outperforms PRS-PGx-CT (which does not consider the impact of LD information), we observed an opposite pattern in our simulations. This is likely because Lasso-based methods are sensitive to the noise, and suffer most when the signal-to-noise ratio is small, which was the case in our simulation data. Finally, for all the methods, the prediction accuracy decreased as the number of causal variants increased given a fixed heritability. This is because, as more causal SNPs were in LD (as a result of more causal SNPs being randomly sampled across the genome) and their effect sizes declined, it became increasingly difficult to

distinguish real signals from noise. Furthermore, we compared $R^2$ of different methods in the treatment and control arms, respectively (Fig. 2c, d). The performance in the treatment arm held a similar pattern as to the $R^2$ in two arms. However, the results from the control arm showed a different pattern. PRS-Dis-LDpred2 seemed to be superior to PRS-PGx methods. Note that under the control arm, the underlying true model becomes $E\mathbf{Y} = \mathbf{G}\beta$. Therefore, a large-scale disease GWAS is able to perfectly recover $\beta$'s with $\hat{\beta}$'s, which implies that the disease PRS ($= \sum_{i=1}^{m} \hat{\beta}_i \mathbf{G}_i$) is able to capture the prognostic effect under control arm. In such condition, disease PRS may show advantage to PGx PRS since it is constructed from a much larger sample size from disease GWAS. Fortunately, our proposed PRS-PGx-Bayes was still comparable to PRS-Dis-LDpred2 (Fig. 2d).

In addition to prediction accuracy, we summarized the predictive $p$-values (i.e., the significance of $S_{pred} \times T$ interaction) across different methods in Fig. 2b. As expected, PRS-PGx methods showed a clear advantage to the PRS-Dis method PRS-Dis-LDpred2. This is not surprising since the disease PRS can fully capture the predictive effect only when the strong assumption (9) is satisfied as we discussed before. Furthermore, our proposed Bayesian approach PRS-PGx-Bayes generally outperformed other methods, which was consistent with our previous observations in terms of $R^2$. P-values, obtained by the two-sided LRT of $S_{PGx}$ from $Y \sim S_{PGx}$ under two arms, respectively, were provided in Supplementary Fig. 2.

As shown in Supplementary Fig. 3, we further compared distributions of $(\hat{\beta}, \hat{\alpha})$ estimated from different PRS-PGx methods versus

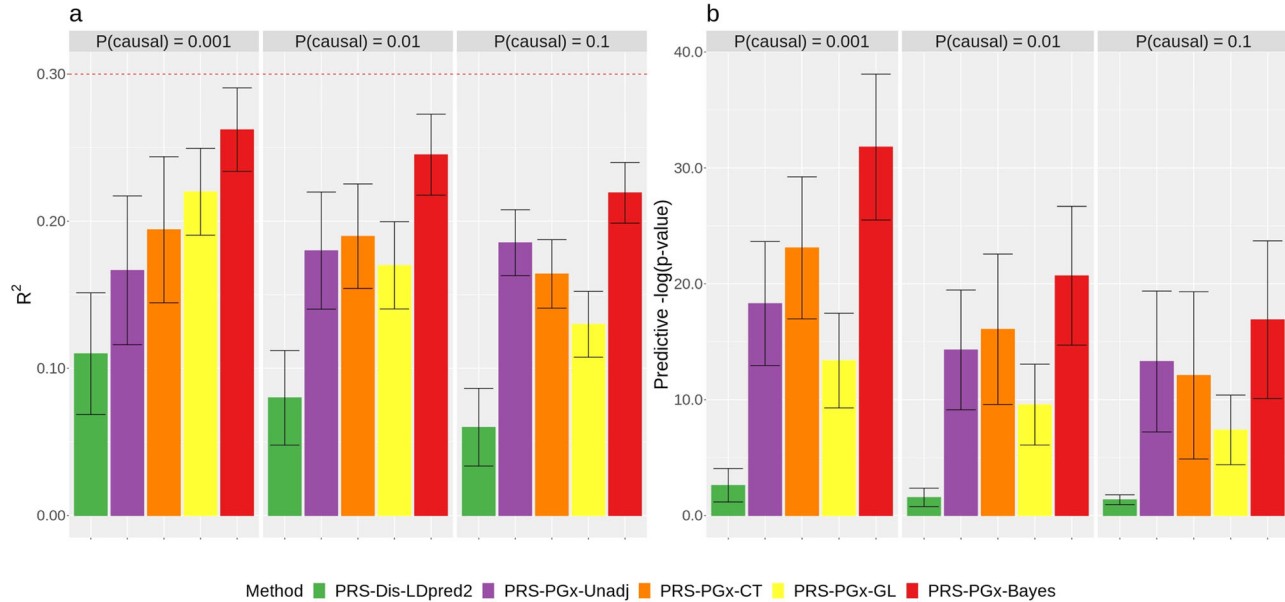

**Fig. 3 | The drug response prediction performance comparison among five methods based on the simulated data with completely separate prognostic and predictive SNPs and heritability fixed at 0.3.** The training sample size for PGx PRS approaches was fixed to be 3000. Numbers of the causal variants for P(causal) = 0.001, 0.01, and 0.1 are 5, 50, and 500, respectively. The performance was assessed in terms of **a** prediction accuracy $R^2$ of $S_{PGx}$ in two arms, **b** predictive $p$-value for the two-sided $S_{pred} \times T$ interaction test. Data are presented as mean values +/− standard deviations (error bars) with 10,000 replications, where results were calculated from the testing sets.

the true value of $(\beta, \alpha)$ under four different genetic architectures, from no signal (P(causal) = 0), sparse signals (P(causal) = 0.001), to dense signals (P(causal) = 0.01 or 0.1). In a scenario when no causal variants were simulated (i.e., null hypothesis scenario), PRS-PGx-CT misidentified a few SNPs. However, the type I error rate was still well controlled. More specifically, PRS-PGx-CT misidentified 7 SNPs, which is comparable to the expected number of false positives (i.e., 5000 × 0.001 = 5). Both PRS-PGx-GL and PRS-PGx-Bayes shrank prognostic and predictive effect sizes to zero. In this scenario when 5, 50, or 500 causal variants were simulated, PRS-PGx-Bayes more accurately estimated the genetic effects compared with the other methods.

To further assess the impact of different implementation strategies, we also performed additional simulations where PRS-PGx-Bayes function was applied on LD blocks jointly (i.e., full LD matrix across LD blocks was used). The simulation settings remained the same as described in Fig. 2. As a sensitivity analysis, we also applied PRS-PGx-Bayes method to the uniform blocks with number of variants in each block as 200, 500, and 2500, respectively. Supplementary Fig. 4 shows that there is a slightly decreasing trend in $R^2$ and $-\log(p\text{-value})$ from using the full genotype matrix to using uniform blocks with size 200. However, such differences across different types of blocks are limited, especially between LD blocks and full genotype matrix. For example, when P(causal) = 0.001, compared to using the full genotype matrix, the LD block approach only decreases $R^2$ by 0.4%. The relative decreases of LD block approach compared to the full genotype approach was summarized in Supplementary Table 1. The table shows that the relative decreases of the LD block approach in simulation studies are very small (i.e., all ≤1.1%).

To assess the performance of the proposed methods under different heritabilities, we conducted sensitivity analyses by setting $H^2 = 0.1$ and 0.5 and the results showed a very similar pattern (Supplementary Fig. 5). Supplementary Fig. 6 shows the sensitivity analysis results when the treatment effect $\beta_T$ was set to 1. The methods performed very similar to when $\beta_T$ was set as 0. Furthermore, we assessed the impact of different scales of the prognostic and predictive effect sizes on the methods' performance. When the two effect sizes were set with different scales ($\psi/\xi = 16$ or 1/16), the sparse group Lasso-based

method (PRS-PGx-SGL) performed the best among the three penalized regression methods, while PRS-PGx-Bayes still outperformed all the other methods (Supplementary Fig. 7). In addition, when $\psi/\xi = 16$ (i.e., the heritability is mostly explained by the prognostic effect), it was not surprising that PRS-Dis-LDpred2 was at least comparable to most PRS-PGx methods in terms of the prediction accuracy $R^2$. But still, the performance of the PRS-Dis methods was much worse than PRS-PGx methods in terms of capturing the predictive effect.

**The constant ratio assumption of PRS-Dis is strongly violated.** We simulated completely separate sets of prognostic and predictive SNPs so that no SNPs were both prognostic and predictive. Under this condition, the ratio of main to interaction effects (i.e., $\beta/\alpha$) for each causal variant was either 0 or ∞. The details of data generation are provided in the "Methods" section.

The simulation results are summarized in Fig. 3, which shows that, when the assumption (11) is not satisfied, the PRS-PGx methods uniformly outperformed PRS-Dis methods in terms of the prediction accuracy across different settings of causal variants. Figure 3a shows that the average $R^2$ of the PRS-Dis methods are all below 0.1 while it is larger than 0.13 for the PRS-PGx methods. Such advantage was even more pronounced for capturing the predictive effect, which was measured by the predictive $p$-value from the two-sided likelihood ratio test of $S_{pred} \times T$ interaction. Figure 3b shows the PRS-Dis-LDpred2 method generates the geometric mean $p$-values > 0.01, but the geometric mean $p$-values of PRS-PGx methods are <1e−8. The detailed results from the three disease PRS methods and the three penalized regression methods are summarized in Supplementary Fig. 8.

**Computational time.** To assess the computational burden of the proposed method, we applied the PRS-PGx-Bayes function with 1000 MCMC iterations to chromosome 6, LD block 33 (the largest LD block with 11,769 SNPs). As a sensitivity analysis, we also explored scenarios by randomly choosing 1000, 3000, 5000, 7000, 9000 SNPs from that block. The real genetic data was obtained from the IMPROVE-IT trial with a sample size of 5661. The effect sizes and phenotype data were simulated with heritability fixed at 0.3, $\psi/\xi = 1$, and P(causal) = 0.01.

**Table 1 | IMPROVE-IT PGx GWAS data analysis results: $R^2$, $p$-values of two-sided test, and effect sizes**

| PRS method | Two arms[a] $R^2$ | T arm[b] $R^2$ | C arm[c] $R^2$ | Two arms $Pval_{G \times T}$ | T arm $Pval_G$ | C arm $Pval_G$ | Two arms $\widehat{\beta}_{G \cdot T}$ (SE) | T arm $\widehat{\beta}_G$ (SE) | C arm $\widehat{\beta}_G$ (SE) |
|---|---|---|---|---|---|---|---|---|---|
| PRS-Dis-CT | 0.165 | 0.152 | 0.191 | 0.041 | 3.0e−09 | 5.1e−17 | −0.031 (0.015) | −0.066 (0.011) | −0.035 (0.004) |
| PRS-Dis-LDpred2 | 0.174 | 0.165 | 0.201 | 0.033 | 4.3e−13 | 6.1e−23 | −0.037 (0.017) | −0.079 (0.011) | −0.042 (0.004) |
| PRS-PGx-Unadj | 0.165 | 0.180 | 0.121 | 0.028 | 1.2e−13 | 2.7e−03 | −0.061 (0.028) | −0.082 (0.011) | 0.057 (0.019) |
| PRS-PGx-CT | 0.184 | 0.241 | 0.070 | 0.009 | 1.7e−15 | 0.01 | −0.095 (0.036) | −0.104 (0.013) | −0.040 (0.016) |
| PRS-PGx-GL | 0.181 | 0.203 | 0.123 | 0.014 | 7.4e−15 | 1.8e−03 | −0.076 (0.031) | −0.093 (0.012) | 0.112 (0.036) |
| PRS-PGx-Bayes | 0.214 | 0.277 | 0.194 | 5.4e−05 | 3.8e−21 | 1.0e−17 | −0.131 (0.032) | −0.124 (0.013) | 0.198 (0.023) |

[a]Two-arm model: $Y \sim T + S_{prog} + T \times S_{pred}$.
[b]T-arm model: $Y \sim S_{PGx}[= Y \sim (S_{prog} + S_{pred})]$, where $S_{PGx} = S_{prog} + T \times S_{pred}$.
[c]C-arm model: $Y \sim S_{PGx}[= Y \sim S_{prog}]$, where $S_{PGx} = S_{prog} + T \times S_{pred}$.

The tuning parameters were selected via cross-validation. The computation was completed on a single core of 2.4 GHz Intel Core i5. We summarized the result in Supplementary Fig. 9, which shows that the computational time increased at the rate of $m^2$ to $m^3$, where $m$ denotes the number of variants. The result also shows that it took roughly 5.9 hours for the largest LD block and 1 h for the median-size LD block (Supplementary Fig. 10) to complete the computation. In practice, since the computation in each LD block is independent, we could further shorten the computational time by parallel computing 1725 LD blocks across the whole genome. In the authors' High Performance Computing working environment, the IMPROVE-IT whole genome analysis took about 35 h, where typically 50 jobs were run simultaneously.

**Polygenic prediction of drug responses in the IMPROVE-IT PGx GWAS study**

We applied the four proposed PRS-PGx methods (PRS-PGx-Unadj, PRS-PGx-CT, PRS-PGx-GL, PRS-PGx-Bayes) and the other two PRS-Dis methods (PRS-Dis-CT and PRS-Dis-LDpred2) to the IMPROVE-IT PGx GWAS summary statistics data to predict the low-density lipoproteins cholesterol (LDL-C) log-fold change at 1-month from the two treatment arms. The two treatment arms are the treatment arm with the combined therapy (Ezetimibe + Simvastatin: 10 mg + 40 mg) and the active control arm with monotherapy (Simvastatin: 40 mg). We adjusted for the age, gender, prior lipid-lowering (PLL) therapy, early glycoprotein IIb/IIIa inhibition in non-ST-segment elevation acute coronary syndrome (EARLY ACS) trial, high-risk ACS diagnosis, baseline LDL-C level, and five top principal components when generating the IMPROVE-IT summary statistics data for the LDL-C drug response phenotype.

To apply PRS-PGx methods to the IMPROVE-IT data, we used nested cross-validation. More specifically, the IMPROVE-IT data was split into five folds in the outer layer of cross-validation with four for training and one for testing. The training set was used to obtain the PGx GWAS summary statistics. In the inner layer of cross-validation, the training set was further split into four folds, three for training and one for validation, to select the optimal tuning parameters (i.e., $p$-value cutoff for PRS-PGx-CT, penalty $\lambda$ for PRS-PGx-GL and $(v, \phi)$ for PRS-PGx-Bayes). We compared performance across different methods with the results summarized from the testing set. The prediction accuracy was measured by $R^2$ and summarized in Table 1. The capabilities of the PRS methods in capturing the prognostic and predictive effects were measured by their effect sizes, as well as association $p$-values, and shown in this table as well.

Consistent with previous simulation results, the two PRS-Dis methods performed poorly in terms of $R^2$ and predictive p-value. In contrast, the PRS-PGx approaches demonstrated an overall improvement in both metrics. For example, PRS-PGx-Bayes increased the prediction accuracy $R^2$ to 0.214 in both arms while compared with 0.174 from the best disease PRS method PRS-Dis-LDpred2. In the treatment arm, PRS-PGx-Bayes improved the $R^2$ by 0.112 while compared with PRS-Dis-LDpred2 (0.277 vs. 0.165). On the other hand, the PRS-Dis method PRS-Dis-LDpred2 was superior to PRS-PGx methods in terms of $R^2$ under the control arm, which might be partially due to the fact that we used the disease GWAS statistics data with much larger sample size ($n > 300,000$) for constructing the disease PRS in the PRS-Dis analyses. But our proposed method PRS-PGx-Bayes still provided a comparable $R^2$ prediction performance (i.e., 0.194 from PRS-PGx-Bayes vs. 0.201 from PRS-Dis-LDpred2). In terms of predictive $p$-value, PRS-PGx-Bayes yielded a much more statistically significant predictive (or interaction) $p$-value 5.4e−05 while compared to 0.033 from PRS-Dis-LDpred2. In addition, $p$-values obtained by the LRT of $S_{PGx}$ showed very similar pattern as the $R^2$ in either treatment or control arm. Table 1 also shows that the marginal effect sizes $\widehat{\beta}_G$ of $S_{PGx}$ from the model $Y \sim S_{PGx}$ under treatment arm were all negative across different PRS-Dis and PRS-PGx methods, indicating that a larger PRS would result in more LDL reduction after 1-month treatment of Ezetimibe + Simvastatin. In the meantime, PRS-PGx-Bayes method outperformed the others with the largest absolute value of effect size $\widehat{\beta}_G$. Similarly, the interaction effect sizes $\widehat{\beta}_{G \times T}$ of $S_{pred}$ from the model $Y \sim T + S_{prog} + T \times S_{pred}$ were all negative across all methods, implying that a larger predictive score would result in a larger treatment effect (i.e., Ezetimibe + Simvastatin combination vs. Simvastatin monotherapy). PRS-PGx-Bayes method is also superior to the others with the largest absolute value of effect size $\widehat{\beta}_{G \times T}$.

We further compared the patient stratification performance across different methods with the results summarized in Fig. 4. In Fig. 4a, we used four fixed quantiles (0–25%, 25–50%, 50–75%, and 75–100%). The results indicated that although overall the population had a positive treatment effect (i.e., Simva+EZ is better), the treatment effects varied across different patient subgroups when stratified by the predictive score. Furthermore, the predictive score determined by PRS-PGx-Bayes was generally superior to other methods for patient stratification. Specifically, ratios of top 75–100% subgroup to bottom 0–25% subgroup in terms of treatment effects were 1.27, 1.48, 1.65, 3.27, 1.94, and 10.28 for PRS-Dis-CT, PRS-Dis-LDpred2, PRS-PGx-Unadj, PRS-PGx CT, PRS-PGx-GL, and PRS-PGx-Bayes, respectively. In Fig. 4b, patients were stratified into top 10%, 20%, ⋯ , 90% percentile of the predictive score vs. the rest, respectively. The corresponding between group differential treatment effect was calculated. Among the six methods, PRS-PGx-Bayes had the largest differential treatment effect across different cutoff points followed by PRS-PGx-CT and PRS-PGx-GL and the rest three methods had the lowest differential treatment effects. The optimal cutoff point for PRS-PGx-Bayes occurred between 50% and 60%, with differential treatment effect around 0.52. Instead of using fixed quantiles, we also determined the optimal quantile cutoffs with the largest differential treatment effect estimated from the 5-fold cross-validation (training and testing) procedures. The corresponding ability of PRS-PGx-Bayes to stratify patients with greater clinical benefits was assessed in different validation sets with the results summarized in Supplementary Fig. 11. The differences in treatment effects

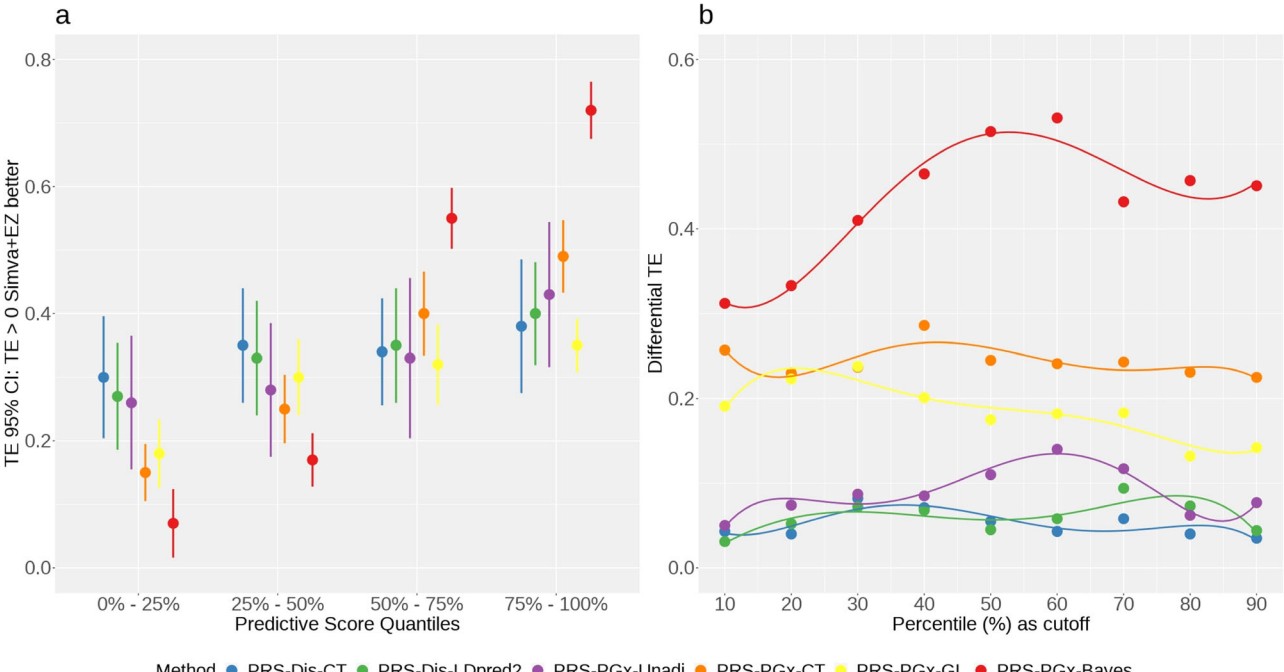

**Fig. 4 | Patient stratification performance of six polygenic prediction methods in the IMPROVE-IT PGx real data analysis with n = 5661 unrelated European samples. a** Quantile plot of treatment effect using four fixed quantiles (0–25%, 25–50%, 50–75%, and 75–100%). Each dot stands for the observed Treatment Effect (TE), and each bar denotes the 95% Confidence Interval (CI). **b** Differential treatment effect when patients were stratified into top 10%, 20%, ⋯ , 90% percentile of the predictive score vs. the rest, respectively.

between high and low predictive score subgroups were very clear in the overall population as well as in four out of five CVs.

In terms of variant selection, PRS-Dis-CT identified 280 SNPs to construct disease PRS with the optimal $p$-value cutoff determined as 5e−08 and PRS-PGx-CT selected 91 SNPs into the PGx PRS with the optimal $p$-value cutoff selected as 1e−07. The number of SNPs identified by PRS-PGx-CT is much smaller because the sample size of the IMPROVE-IT PGx GWAS data is much smaller than that from the disease GWAS summary statistics data. For PRS-PGx-Bayes, it tends to shrink effect sizes of most SNPs close to zero (but not exactly equivalent to zero), which is consistent with our simulation results. Distributions of the predictive effect sizes of the whole genome SNPs estimated by PRS-PGx-CT, -GL, and -Bayes are shown in Supplementary Fig. 12. The corresponding information of the top 20 SNPs with the largest predictive effect sizes (>0.1) reported by PRS-PGx-Bayes was summarized in Supplementary Table 2, most of which were with the previous association evidence from literature and the Open Targets database (https://genetics.opentargets.org/).

## Discussion

In this article, we develop a series of PRS-PGx methods to construct PGx-based PRS and predict the polygenic component of drug responses in PGx studies. To our best knowledge, no existing methods can be directly applied to jointly model both prognostic and predictive effects for drug response prediction. The necessity of using PGx PRS approaches instead of disease PRS approaches is validated by the proof of extremely stringent assumptions needed for the disease PRS approach to predict drug response. Our proposed PRS-PGx methods include a simple method using whole genome variants (PRS-PGx-Unadj), a clumping and $p$-value thresholding method (PRS-PGx-CT), three penalized regression methods (PRS-PGx-L, PRS-PGx-GL, PRS-PGx-SGL), and a novel Bayesian regression method (PRS-PGx-Bayes). Except for the penalized regression-based algorithms, all the other methods can take PGx summary statistics as input without relying on individual-level genotype and phenotype data. Compared with the

PRS-Dis methods, the PRS-PGx approaches can shrink variants' main and interaction effect sizes simultaneously, and construct PGx scores including a prognostic PRS and a predictive PRS. Thus, in our PRS-PGx-Bayes method, we propose to accommodate both effects and their correlation by modeling a variance-covariance matrix. Although the inverse Wishart (IW) prior is widely used, in this paper, we choose to use the hierarchical half-t prior[27] instead due to the limitation of IW (i.e., IW prior imposes a dependency between the correlations and the variances). Moreover, by introducing global-local continuous shrinkage priors on SNP effect sizes, our proposed PRS-PGx-Bayes method is more robust to varying relationships between main and interaction effects compared to other PRS-PGx methods. Our extensive simulation studies and the PRS analyses based on the IMPROVE-IT PGx GWAS data show that the proposed PRS-PGx methods generally outperform the PRS-Dis methods in terms of the prediction accuracy $R^2$ and the significance of their predictive effects. Furthermore, the PGx PRS Bayesian approach (PRS-PGx-Bayes) is superior to all the other methods, under different genetic architectures. Interestingly, we find that the C+T method (PRS-PGx-CT) can outperform the penalized regression-based methods (PRS-PGx-L, PRS-PGx-GL, and PRS-PGx-SGL) in some of the scenarios. This pattern was also observed in Mak et al.[11]. Our results suggest that Lasso-based methods are sensitive to the noise, and perform poorly when the signal-to-noise ratio is small (Supplementary Figs. 5 and 7). Further study is needed to examine the difference more comprehensively.

Despite the successful application of polygenic risk score in disease genetic studies for disease prediction and stratification, the PRS analyses in PGx studies from randomized clinical trials are usually more complex. On one hand, although sample sizes from PGx studies are usually smaller than those in disease GWAS, the significantly larger effect sizes from PGx studies[28,29] can result in decent power to detect variants associated with drug response phenotypes and further enable good drug response prediction performance of PGx PRS. On the other hand, the availability of summary statistics from PGx GWAS is expected to increase quickly (i.e., similar as the case for the availability of

summary statistics from disease GWAS). Therefore, statistical methods customized for PGx PRS analysis are urgently needed for drug response prediction and patient stratification in PGx GWAS studies, with the ultimate goal of achieving precision medicine. Unfortunately, there are currently very limited efforts of constructing PGx PRS and successfully applying to efficacy-based PGx studies[16,22,23]. The most popular practice of PRS analysis strategies in literature is to construct PRS from disease GWAS and apply to PGx data. We used rigorous statistical modeling to prove that PRSs built from disease GWAS lack the prediction potential in reaching the full heritability of drug responses. Qualitatively speaking, the disease PRS approach is only able to fully predict drug responses under an extremely strong assumption that all variants used for constructing PRS have the same relationship between their main effect and interaction effect (i.e., the ratio between the prognostic G and predictive $G \times T$ effect sizes is a constant). The violation of such assumption may result in very limited predictive power. For example, we showed in our theoretical analysis that any PRS developed from disease GWAS can explain at most 46% variability of the LDL-C drug response in the IMPROVE-IT PGx GWAS data. In addition, the disease PRS likely lacks the molecular specificity to be directly informative for clinical interventions[30] since the PRS is built from variants with only prognostic effects (i.e., without considering the variants with predictive effects related to treatment). This may partially explain why there has been little investigation into PGx PRS and very few successful PGx PRS applications have been published for predicting drug responses in the PGx space so far. One successful example using an analysis strategy most closely resemble our PGx PRS analysis strategy is from Lanfear et al.'s paper[16]. A direct PGx polygenic response predictor (PRP) was constructed from a genome-wide analysis of $\beta$-Blocker (BB) x SNP interaction and successfully predicted all-cause mortality (BB survival benefit) in European ancestry patients with reduced ejection fraction heart failure[16]. The authors built the PRS with the selected variants (using the simple p-value thresholding method) with only predictive effects (i.e., from 44 SNPs with strong BB x SNP interaction) and then applied to the validation PGx GWAS data for PRS effect evaluation and patient stratification. Compared with the PGx PRS construction in this example, the PGx PRS analysis strategy embedded in our PRS-PGx methods constructs both the prognostic and predictive PRSs by jointly modeling both effects in the base GWAS data and then tests them on validation data. This provides a systematic way to understand the full picture of PRS' association with drug response. Furthermore, in addition to the p-value thresholding method, the more advanced (Bayesian regression based) method PRS-PGx-Bayes can also be applied to such application examples, which has been demonstrated to outperform the simple *p*-value thresholding method in our simulations and real data analyses. In summary, our drug response prediction analysis strategies based on proposed PRS-PGx methods, especially PRS-PGx-Bayes, are highly recommended over the various strategies and existing methods from current literature.

The genetic architectures of responses from a single drug or drug classes determine the proportion of variants across the human genome contributing to their heritability and the proportion of variants with small-, moderate- or large-effects in capturing the heritability. Our research tackles the challenges of PRS methodology in PGx so that PRS can be directly and accurately used for drug response prediction and patient stratification. However, it is usually challenging to find two independent PGx GWAS data or a summary statistics data from a relatively large PGx GWAS study with the trait same or similar as that in an independent testing PGx GWAS study. In addition to discovering the top variants with large effects associated with drug responses (like what most current PGx GWAS do), we call for also carefully examining genetic architectures of drug responses in PGx GWAS and sharing summary statistics data from those studies in the public domain.

Furthermore, recording both the prognostic and predictive (the genotype by treatment interaction) effects (β, α) as GWAS summary statistics in future PGx studies from randomized clinical trials and sharing them will enable the wide application of the proposed PRS-PGx methods and accelerate the development and deployment of PRS based on PGx data for precision medicine.

Recent studies[31] have shown that Genotype-by-Environment Interaction (GEI or G × E) may explain a significant proportion of phenotypes compared with the main genotype (G) test in disease genetics studies. Our proposed PRS-PGx methods can be directly applicable to the scenario of genome-wide GEI test or G + G × E joint test if environmental variables are considered in the PRS development and deployment in disease genetics. In addition, although we mainly demonstrate the PRS-PGx methods in the analysis of continuous drug response phenotypes, all of them can be directly applied to binary (i.e., drug-induced adverse reactions or other drug safety responses) and survival (i.e., time-to-event drug responses) phenotypes except for PRS-PGx-Bayes. It is straightforward to extend the PRS-PGx-Bayes to analyzing binary and survival endpoints by adopting the Bayesian logistic regression[32] and Bayesian Cox proportional hazards model[33–35] instead of the Bayesian linear regression. How to derive their posterior probabilities remains a question for future research. Moreover, our methods are developed based on single-ethnic population (e.g., European population). A direct application of this score to other ethnic groups may result in considerable loss in prediction accuracy. With the rapid growth of non-European genomic resources in recent years, it is, therefore, of great both research and public interests to extend the proposed PRS-PGx approaches to the trans-ethnic scenario in future. In addition, for the purpose of effect size shrinkage, the currently existing Bayesian methods lack a systematic way to determine the optimal prior. For example, the LDpred method uses the spike-and-slab prior[12]; the SBayesR method uses the spike-and-slab prior by replacing normal with mixture normal[36]; the DPR method uses the Dirichlet process prior[37]; and Griffin and Brown use the Normal-Gamma shrinkage prior[38]. The PRS-CS method[14] and our proposed PRS-PGx-Bayes method use one of the most popular continuous shrinkage priors (global-local scale mixtures of normals, i.e., the Horseshoe prior). One potential drawback of the Horseshoe prior is that there has been no consensus on how to place a prior for $\phi$ based on the information about the sparsity[39], neither by grid searching for the optimal $\phi$ (as in PRS-PGx-Bayes), nor by applying full Bayesian inference (as in PRS-CS). Thus, another possible future research direction is to systematically study the impact of different priors on PRS-PGx-Bayes performance and then determine the optimal one. Furthermore, if the Horseshoe prior is used, we may further improve the PRS-PGx-Bayes algorithm by automatically updating $(v, \phi)$ based on the sparsity information instead of using grid searching for the tuning parameters. Last, we suggest applying our proposed methods by LD blocks. Expanding the size of blocks (e.g., using the whole chromosome) may slightly improve prediction accuracy but also significantly increase computational costs. On the other hand, further reducing the size of blocks (e.g., using the uniform blocks with smaller size) can reduce run-time but also possibly increase the bias by missing long-range LD. We believe that the by LD block strategy is a right trade-off between decent modeling accuracy and feasible computational time. We also acknowledge that future work is needed to further improve the computational efficiency to incorporate more SNPs for simultaneous analysis.

As next-generation sequencing and other genetic platforms become much cheaper and more routinely embedded within PGx research studies, more rare variants may be evaluated and discovered to associate with drug responses. There will be an increasing need to consider pharmacogenomic variants, both common and rare with either large, moderate, or small effects to predict patients' drug

responses. The PRS-PGx methods we develop are promising in advancing precision medicine by improving drug response (efficacy and/or safety) prediction in PGx studies. Our efforts of developing the PRS-PGx methods, which identify optimal ways for PRS construction by jointly modeling the prognostic and predictive effects, is an important step for accelerating the translation of PRS to clinical practice.

## Methods

The four PRS-PGx methods (PRS-PGx-Bayes, PRS-PGx-Unadj, PRS-PGx-CT, and PRS-PGx-L, -GL, -SGL) we propose are described in this section. We leave the brief description of the PRS-Dis-LDpred2 method to Supplementary Method C and Supplementary Fig. 13 since it is an existing method. The overview of PRS-Dis and PRS-PGx methods is summarized in Supplementary Table 3. The workflow and details of simulation studies and real data analyses are also discussed.

### PRS-PGx-Bayes

Recall that the Bayesian regression model has been specified in Eq. (1), and we assume $\mathbf{Y}$ and $\mathbf{G}$ have been standardized. Furthermore, we assume the residual variance $\sigma^2$ follows a non-informative scale-invariant Jeffreys prior, that is, $p(\sigma^2) \propto \sigma^{-2}$ as suggested by Ge et al.[14] Regarding the priors of effect sizes, we extend the idea of global-local scale mixtures of normals[14] to the two-dimensional scenario as indicated in Eq. (2):

$$\begin{pmatrix} \beta_j \\ \alpha_j \end{pmatrix} | \sigma^2, \phi, \psi_j, \xi_j, \rho_j \sim \text{MVN}\left(\mathbf{0}, \frac{\sigma^2}{n}\phi M_j\right), \text{ where } M_j = \begin{bmatrix} \psi_j & \rho_j\sqrt{\psi_j\xi_j} \\ \rho_j\sqrt{\psi_j\xi_j} & \xi_j \end{bmatrix} \sim g,$$

where $\phi$ is a global scaling parameter that is shared across all effect sizes; $\psi_j$ and $\xi_j$ are local and marker-specific scaling parameters; $\rho_j$ is the marker-specific correlation between the two effect sizes $\beta_j$ and $\alpha_j$; and $g(\cdot, \cdot)$ is an absolutely continuous and two-dimensional mixing density function.

We first note that, given the prior information $\sigma^2$, $\phi$, and $M_j$, $j = 1, 2, \cdots, m$, and the summary statistics (i.e., the effect size estimates) $\hat{\mathbf{b}} = \mathbf{X}'\mathbf{Y}/n$ from PGx GWAS, the posterior mean of $\mathbf{b}$ is

$$E[\mathbf{b}|\hat{\mathbf{b}}] = (\mathbf{D} + \Omega^{-1})^{-1}\hat{\mathbf{b}}, \tag{12}$$

where

$$\Omega = \begin{bmatrix} \Psi & P \\ P & \Xi \end{bmatrix}, \Psi = \text{diag}(\phi\psi_j), \Xi = \text{diag}(\phi\xi_j), P = \text{diag}\left(\phi\rho_j\sqrt{\psi_j\xi_j}\right),$$

and

$$\mathbf{D} = \mathbf{X}'\mathbf{X}/n = \text{cor}([\mathbf{G} \quad \mathbf{G} \times \mathbf{T}])$$

can be obtained from LD reference panel as illustrated in Supplementary Method D and Supplementary Fig. 14.

To provide further insights on Eq. (12), we consider several scenarios. First, assume $\psi_j \equiv 1$, $\xi_j \equiv 1$, and $\rho_j \equiv 0$, Eq. (12) is equivalent to the Ridge regression and all effect sizes are shrunk towards zero at the same constant rate controlled by the global shrinkage parameter $\phi$ (i.e., the penalty). Second, assume a one-to-one treatment-placebo allocation ratio ($\mu_T = 0.5$), unlinked genetic markers ($\sigma_{ij} \equiv 0$ for $i \neq j$) and $\rho_j \equiv 0$ (i.e., within each SNP, the two effect sizes are independent). We can derive the formulas explicitly for $E[\beta_j|\hat{\beta}_j]$ and $E[\alpha_j|\hat{\alpha}_j]$ (Supplementary Method E). Under the simplified scenario where all markers'

MAFs are small, $f_j \equiv f \to 0$, we can show that:

$$E[\beta_j|\hat{\beta}_j] \approx \frac{\hat{\beta}_j - \frac{c}{s_j}\hat{\alpha}_j}{t_j - c^2/s_j},$$

$$E[\alpha_j|\hat{\alpha}_j] \approx \frac{\hat{\alpha}_j - \frac{c}{t_j}\hat{\beta}_j}{s_j - c^2/t_j},$$

where $t_j = 1 + \phi^{-1}\psi_j^{-1}$, $s_j = 1 - f + \phi^{-1}\xi_j^{-1}$, and $c = \sqrt{(1-f)/2}$. To understand the above equations, we can interpret $1/t_j = \frac{1}{1+\phi^{-1}\psi_j^{-1}}$, $1/s_j = \frac{1}{1-f+\phi^{-1}\xi_j^{-1}}$ as the shrinkage factors. Therefore, $\hat{\beta}_j/t_j$ and $\hat{\alpha}_j/s_j$ are the 'shrunk' effects: $t_j = s_j = 1$ indicates no shrinkage while $t_j = s_j \to \infty$ yields full shrinkage. The correlation $c$ between $G_j$ and $G_j \times T$ also contributes to the second part of the numerator because the bias induced by the positive correlation needs to be corrected.

In the PRS-PGx-Bayes method, $M_j$ denotes the variance-covariance matrix. It is a common practice to use an inverse Wishart (IW) distribution as the conjugate prior for the covariance matrix of a multivariate normal distribution. However, the IW prior has its own limitations. The IW prior imposes a dependency between the correlations and the variances: larger variances automatically imply the absolute value of the correlation near one while small variances imply the correlation near zero[40,41]. Therefore, in this study, we propose to use the hierarchical half-t prior[27], which is more flexible than the IW prior by adding the degrees of freedom parameter to the scale matrix. Specifically, we assume

$$M_j \sim W^{-1}(B_j, 2\nu+1), \text{ where } B_j = 4\nu\begin{bmatrix} \delta_j & 0 \\ 0 & \lambda_j \end{bmatrix}, \delta_j \sim G(b_1, 1), \lambda_j \sim G(b_2, 1),$$

(13)

where $W^{-1}(B_j, 2\nu+1)$ denotes the inverse Wishart distribution with scale matrix $B_j$ and $\nu$ degrees of freedom and G is a Gamma distribution. Equation (13) implies marginal distributions of the variances and the correlation as:

$$\psi_j \sim \text{iG}(\nu, 2\nu\delta_j), \quad \xi_j \sim \text{iG}(\nu, 2\nu\lambda_j), \quad p(\rho_j) \propto (1-\rho_j^2)^\nu,$$

where iG denotes the inverse Gamma distribution. By using this definition, changing the correlation does not necessarily result in a change in the variances, which are instead determined through $\delta_j$ and $\lambda_j$.

In practice, by using LD information from an external reference panel (i.e., 1000 Genomes data), the method can be applied to PGx GWAS summary statistics and does not require individual-level genotype and phenotype data. PRS-PGx-Bayes approach updates $\mathbf{b} = (\beta, \alpha)$, $\sigma^2$, $(\psi, \xi, \rho)$, $(\delta, \lambda)$ sequentially based on their posterior distributions (as described in Supplementary Method E), where we set $b_1 = b_2 = 1/2$ as suggested by Ge et al.[14] Also as proposed by Ge et al.[14], to avoid numerical issues caused by collinearity between SNPs, we set $\phi\psi_j \leq \rho$ and $\phi\xi_j \leq \rho$, where $\rho = 1$ is a constant number. In addition, we partition the genome into 1725 largely independent genomic regions[42] (https://bitbucket.org/nygcresearch/ldetect-data/src/master/) estimated using data from the 1KG European sample, and further conduct multivariate update of the effect sizes within each LD block. The distributions of block sizes in 1KG and IMPROVE-IT data are summarized in Supplementary Fig. 10 a and b, respectively. The largest LD block (chr 6, block 33) after matching IMPROVE-IT PGx data to 1KG contains 11,769 SNPs in total. The same strategy is also applied to the other methods. As shown in Supplementary Fig. 4 and Supplementary Table 1, this strategy is justified by that fact that only a small relative difference ($\leq 1.1\%$) is observed when the PRS-PGx-Bayes

function is carried out by LD blocks, compared to across multiple LD blocks in simulation studies. The overall PRS-PGx-Bayes algorithm is described in Algorithm 1.

### Algorithm 1. PRS-PGx-Bayes: Performed within each LD block

**Result:** Estimated $\mathbf{b}^{\text{Bayes}} = (\beta^{\text{Bayes}}, \alpha^{\text{Bayes}})$ after shrinkage

Hyper-parameters: $v$, $\phi$, $b_1 = b_2 = 1/2$;

Summary statistics[a]: PGx GWAS $\hat{\mathbf{b}} = (\hat{\beta}, \hat{\alpha})$, LD reference panel from 1KG;

Initialization: $\psi$, $\xi$, $\rho$, $\sigma^2$, n.itr, n.burnin, n.gap, n.pst = (n.itr - n.burnin) / n.gap;

**while** $t < n.itr$ **do**

 Update $\mathbf{b} = (\beta, \alpha) \sim$ Bivariate Normal;

 Update $\sigma^2 \sim$ Inverse Gamma;

 Update $\psi$, $\xi$, and $\rho \sim$ Inverse Wishart;

 **if** $\psi > 1/\phi$, **then** $\psi = 1/\phi$; **if** $\xi > 1/\phi$, **then** $\xi = 1/\phi$;

 Update $\delta$ and $\lambda \sim$ Gamma;

**end**

$\mathbf{b}^{\text{Bayes}} = (\mathbf{0}, \mathbf{0})$;

**while** $n.burnin < t < n.itr$ and $(t - n.burnin) \%\% n.gap = 0$ **do**

 $\mathbf{b}^{\text{Bayes}} = \mathbf{b}^{\text{Bayes}} + \mathbf{b}^t / \text{n.pst}$;

**end**

**Output:** $\mathbf{b}^{\text{Bayes}}$

[a]The list of the full PGx GWAS summary statistics is provided in Supplementary Method F.

### PRS-PGx-Unadj

The PGx PRS include two parts: prognostic PRS ($S_{\text{prog}}$) and predictive PRS ($S_{\text{pred}}$). The unadjusted PGx PRS is the sum of all genetic markers across the whole genome, weighted by their marginal prognostic and predictive effect size estimates (i.e., $\hat{\beta}$ and $\hat{\alpha}$ from PGx GWAS summary statistics), respectively.

### PRS-PGx-CT

The PRS-PGx-CT method constructs both the prognostic and predictive PRS using the variant LD-clumping and $p$-value thresholding steps, in a similar manner as the disease PRS C+T method. Specifically, in the clumping step, for any pair of SNPs that have a physical distance smaller than 250 kb and an LD $r^2 > 0.01$, the less significant SNP is removed. Furthermore, in the thresholding step, the prognostic and predictive effect size estimates of SNPs, whose 2-df two-sided test (i.e., joint test of G + G × T, obtained from PGx GWAS summary statistics) p-values not passing the threshold $P_T$, will be shrunk to zero. And then the remaining SNPs are kept with both types of effects. We consider $P_T \in \{5e-08, 1e-07, 1e-06, 1e-05, 1e-04, 0.001, 0.01, 0.1, 1\}$ in this paper. The $P_T$ value that produces the highest prediction accuracy in a validation data set is selected, and the predictive performance is assessed in an independent testing set.

### PRS-PGx-L, -GL and -SGL

In the PRS-PGx-L, -GL and -SGL methods, penalized regression is used to solve Eq. (1) with individual-level data. Assuming independence between prognostic and predictive effect sizes within each SNP, a direct solution is to minimize the following equation while using a Lasso framework (PRS-PGx-L based on glmnet R package v4.1.1 https://cran.r-project.org/web/packages/glmnet/index.html):

$$f(b) = \frac{1}{2}\|\mathbf{Y} - \sum_{j=1}^{m} \mathbf{X}_j \mathbf{b}_j\|_2^2 + \lambda\|\mathbf{b}\|_1,$$

where $\mathbf{X}_j = [\mathbf{G}_j, \mathbf{G} \times \mathbf{T}_j]$, and $\mathbf{b}_j = (\beta_j, \alpha_j)$. $\|\cdot\|_1$ and $\|\cdot\|_2$ stand for L1-norm and L2-norm, respectively. Assuming if a SNP is included into the model, both prognostic and predictive effects of that SNP may be non-zero, then Group Lasso[43] (PRS-PGx-GL based on gglasso R package v1.5 https://cran.r-project.org/web/packages/gglasso/index.html) might be

appealing by considering each genetic marker as a group:

$$f(b) = \frac{1}{2}\|\mathbf{Y} - \sum_{j=1}^{m} \mathbf{X}_j \mathbf{b}_j\|_2^2 + \lambda\sum_{j=1}^{m}\sqrt{p_j}\|\mathbf{b}_j\|_2,$$

where $p_j = 2$ denotes the group size. Finally, if we assume sparsity at both the group and individual feature levels, we also consider the Sparse Group Lasso[44] (PRS-PGx-SGL based on SGL R package v1.3 https://cran.r-project.org/web/packages/SGL/index.html) whose penalty is a linear combination of penalties from Lasso and Group Lasso:

$$f(b) = \frac{1}{2}\|\mathbf{Y} - \sum_{j=1}^{m} \mathbf{X}_j \mathbf{b}_j\|_2^2 + \alpha\lambda\|\mathbf{b}\|_1 + (1-\alpha)\lambda\sum_{j=1}^{m}\sqrt{p_j}\|\mathbf{b}_j\|_2.$$

## Simulations

We performed simulation studies using real genetic data from the IMPROVE-IT trial ($n = 5661$ in the PGx subset population) and the 1KG European sample ($n = 503$) as an external LD reference panel. 5000 SNPs were randomly chosen from LD blocks 31 and 32[42] on chromosome 19, which were matched between the IMPROVE-IT and the 1KG datasets. To mimic disease GWAS data, the sample size was increased to $n = 20,000$ via random mating (Supplementary Method G). The SNP prognostic and predictive effect sizes were simulated jointly with the following distribution:

$$\begin{pmatrix} \beta_j^{(k)} \\ \alpha_j^{(k)} \end{pmatrix} \sim \begin{cases} \text{MVN}(0, \Sigma_k) & \text{with probability } \pi_k, \\ 0 & \text{with probability } 1 - \pi_k, \end{cases} \quad (14)$$

where $\pi_k \sim \text{Beta}(\text{P(causal)}, 1 - \text{P(causal)})$, $j$ denotes the $j$-th SNP, and $k$ the $k$-th LD block. Equation (14) implies that proportion of causal variants varies from different LD blocks, but the overall proportion of causal variants across the whole genome is controlled by P(causal). Furthermore,

$$\Sigma_k = \begin{bmatrix} \psi & \rho_k\sqrt{\psi\xi} \\ \rho_k\sqrt{\psi\xi} & \xi \end{bmatrix}, \text{where } \rho_k \sim \text{Uniform}(0, 1). \quad (15)$$

We explored different scenarios when $\psi/\xi = 1$ (i.e., the prognostic effect was in the same scale with the predictive effect); and when $\psi/\xi = 16$ or $1/16$ (i.e., one effect was dominant to the other effect). It is worth noting that Eq. (14) indicates that each causal variant would carry some degree of prognostic effect, and some degree of predictive effect. In addition, to generate completely separated prognostic and predictive SNPs, we randomly chose half of the causal variants and only kept their prognostic effects (i.e., artificially shrank $\alpha_j = 0$); for the rest half of the causal variants, only predictive effects were kept (i.e., $\beta_j = 0$).

Five clinical factors (age, gender, prior lipid-lowering (PLL) therapy, EARLY acute coronary syndrome (ACS) trial, and high-risk ACS diagnosis) were considered as covariates. To mimic the disease GWAS data, the phenotype was generated as

$$\mathbf{Y}_{n \times 1} = \mathbf{X}_{n \times 5}\mathbf{1}_{5 \times 1} + \mathbf{G}_{n \times m}\beta_{m \times 1} + \epsilon_{n \times 1},$$

where $n = 20,000$ and $m = 5000$. To mimic the PGx GWAS data, the simulated trait was generated as

$$\mathbf{Y}_{n \times 1} = \mathbf{X}_{n \times 5}\mathbf{1}_{5 \times 1} + \beta_T\mathbf{T}_{n \times 1} + \mathbf{G}_{n \times m}\beta_{m \times 1} + (\mathbf{G} \times \mathbf{T})_{n \times m}\alpha_{m \times 1} + \epsilon_{n \times 1},$$

where $n = 5661$ and $m = 5000$. In the above two equations, $\epsilon_{n \times 1} \sim \text{N}(\mathbf{0}, \sigma^2\mathbf{I}_n)$, where $\sigma^2$ was determined by the heritability, which was set to 0.1, 0.3, and 0.5.

In each replicate, we randomly chose either 1000 or 3000 patients from PGx GWAS data as the PGx training dataset (to build the

PGx PRS); and the other 1000 patients as the independent testing dataset (to evaluate the predictive performance of all the PRS-Dis and PRS-PGx methods). Specifically, the observed phenotype $Y$ in the testing set was first adjusted by the clinical factors $X$ and the treatment $T$, and we obtained the adjusted phenotype $Y_{adj}$. Then the predictive effect was measured by the $p$-value from the two-sided likelihood ratio test of $S_{pred} \times T$ interaction in the model $Y_{adj} \sim S_{prog} + S_{pred} \times T$. The prediction accuracies were quantified by $R^2$ between the adjusted phenotype and the predicted ones from $Y_{adj} \sim S_{PGx}$ under two arms, the treatment arm, and the control arm, respectively. $P$-values were also obtained by the two-sided LRT of $S_{PGx}$ under treatment and control arms, respectively. The entire workflow of the simulation studies is shown in Supplementary Fig. 15.

### IMPROVE-IT PGx GWAS data analysis

We applied two PRS-Dis methods (PRS-Dis-CT and PRS-Dis-LDpred2) and four PRS-PGx methods (PRS-PGx-Unadj, PRS-PGx-CT, PRS-PGx-GL, and PRS-PGx-Bayes) to the prediction of the drug response (low-density lipoprotein cholesterol log-fold change at 1-month) from the IMPROVE-IT PGx GWAS although LDL-C was measured longitudinally at multiple time points. IMPROVE-IT is a phase 3b, multicenter, double-blind, randomized study to establish the clinical benefit and safety of Vytorin (Ezetimibe/Simvastatin tablet) versus Simvastatin monotherapy in high-risk subjects[24] (clinical trial registry number: NCT00202878). The ethics committee at each participating center approved the protocol and amendments. All IMPROVE-IT trials were carried out in accordance with the Declaration of Helsinki, current guidelines on Good Clinical Practices and local ethical and legal requirements. All participants provided voluntary written informed consent before trial entry. The details of the endpoint, genotyping, genotype QC, and imputation for this GWAS analyses are introduced elsewhere[25]. After GWAS QC and SNP imputation, there were 9,407,967 variants and 6502 subjects are available for analyses. The subjects were further filtered down to 5661 subjects for the GWAS analyses by excluding subjects who had a cardiovascular event prior to month 1, since cardiovascular events prior to this time point may affect LDL-C in a manner unrelated to treatment. A total of 5661 European subjects were included for analysis of the LDL-C endpoint.

For the PRS-Dis approaches (PRS-Dis-CT and PRS-Dis-LDpred2), the LDL-C disease GWAS summary statistics, obtained from the Global Lipids Genetics Consortium Results[45] (http://csg.sph.umich.edu/willer/public/lipids2017/), were used to construct disease PRS in the IMPROVE-IT data (as the independent testing set). For PRS-PGx methods (PRS-PGx-Unadj, PRS-PGx-CT, PRS-PGx-GL, and PRS-PGx-Bayes), due to the lack of independent PGx data for training, we alternatively proposed to use a 5-fold cross-validation to evaluate their performance. More specifically, we split the IMPROVE-IT data into five folds; in each CV step, we chose four of them as the training set, and used the remaining one as the testing set. The PGx GWAS summary statistics, obtained from the training set, were used to construct PGx PRS in the testing set, and only the prognostic and predictive scores of patients in the testing set were recorded. The above procedures were repeated and the PGx PRS for all the patients in the IMPROVE-IT PGx GWAS data (when they served as the testing set) were obtained. Finally, the predictive performance was evaluated in the same criteria as described in the "Simulations" section, as well as the quantile plot for patient stratification.

It is worth noting that in each cross-validation, the GWAS summary statistics data were generated from the training set by running GWAS analysis with the model: $\log \mathbf{Y}_1 - \log \mathbf{Y}_0 = \beta_0 + \beta_{Y_0} \log \mathbf{Y}_0 + \beta_T \mathbf{T} + \mathbf{G}\beta + (\mathbf{G} \times \mathbf{T})\alpha + \mathbf{X}\gamma$ where $\mathbf{Y}_1$ is the on-treatment LDL-C response, $\mathbf{Y}_0$ is the baseline LDL-C response, $\beta$ is the prognostic effect, $\alpha$ is the predictive effect and the covariate matrix $\mathbf{X}$ included age, gender, prior lipid-lowering therapy, early Acute Coronary Syndrome (ACS) trial,

high-risk ACS diagnosis, and the top five principal components. As recommended by Zhang et al.[25], we adjusted for the baseline LDL-C level $Y_0$ (in the log scale) in the model to appropriately control the type I error rate (or genome inflation). The PGx GWAS summary statistics data, including the prognostics and predictive effects ($\hat{\beta}$ and $\hat{\alpha}$), the minor allele frequencies (MAF), the two-sided 2df (G + G × T) test $p$-values, the SNP positions, the standard deviation of response, and the mean of treatment, were further used for the PGx PRS based drug response analyses. Detailed information about the summary statistics is provided in Supplementary Method F.

### Reporting summary

Further information on research design is available in the Nature Research Reporting Summary linked to this article.

### Data availability

MSD's data sharing policy, including restrictions, is available at http://engagezone.msd.com/ds_documentation.php. Requests for access to the PGx GWAS summary statistics results from this IMPROVE-IT clinical study data can be submitted through the EngageZone site or via email to dataaccess@merck.com.

### Code availability

Our methods are implemented in the PRSPGx R package v0.3.0, freely available at the Comprehensive R Archive Network (CRAN): https://cran.r-project.org/web/packages/PRSPGx/index.html.

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

## Acknowledgements

The authors would like to thank the study participants in the IMPROVE-IT trials, Peter M. Shaw and Yiwei Zhang for useful discussion about the IMPROVE-IT data.

## Author contributions

J.S. and H.Z. oversaw the study. The theory underlying PRS-PGx was conceived of and developed by J.S., H.Z., and S.Z. S.Z. developed PRS-PGx software and performed the simulation studies and real data analyses. J.S. provided the IMPROVE-IT GWAS summary statistics data. J.S., H.Z., and S.Z. conducted result interpretation. J.S., S.Z., and H.Z. wrote the first version of the manuscript. D.V.M. also contributed to the writing. All authors provided input and revisions for the final manuscript.

## Competing interests

S.Z., H.Z., D.V.M., and J.S. are employees at Merck Sharp & Dohme Corp., a subsidiary of Merck & Co., Inc., Rahway, NJ, USA.
