## [Peer Review File · Nature Communications]

Pharmacogenomics Polygenic Risk Score for Drug Response Prediction Using PRS-PGx MethodsEditorial Note: Parts of this Peer Review File have been redacted as indicated to remove third-party material where no permission to publish could be obtained.

REVIEWER COMMENTS

Reviewer #1 (Remarks to the Author):

Zhai and colleagues present methods that use GWAS summary statistics and individual-level data from pharmacogenomics settings to build polygenic scores (PGS) with both good prognostic and predictive values. They compare their method with traditional PGS methods that use GWAS summary statistics only. I am not an expert in RCT, but I imagine this could have a major impact as a better use of PGS in RCTs. However, I have several comments and questions I would like the authors to address before I could recommend this for publication in Nature Communications. Overall, I fear that the size of the data used and the implementation of the new methods are not convincing enough.

Major comments (in no particular order):

- I am not sure I understand what are the prognostic effects (betas) in the model used; when we adjust for the baseline level Y_0 in the model, what would be the effects of the genetics on this change (i.e. $Y_1 - Y_0$ or Y_1 / Y_0)?
- PRS-PGx-Bayes seems to be somewhat an extension of the PRS-CS method (using very similar Bayesian setting and tricks like regularizing $\phi \times \psi \leq 1$ that are not really mentioned as borrowed from there); can this really be considered a novel method?
- Implementations of all methods presented here rely on very crude R code, e.g. calling `cor()` on the whole genotype matrix (i.e. computing and storing the full dense LD matrix); this seems highly inefficient and would never scale to the current data sizes used in summary statistics (e.g. even when restricting to e.g. 1M HapMap3 variants).
- On the same note, the correlation/LD matrix between variants can often be considered as sparse (banded or even block-diagonal). This is one of the main reasons PGS methods can be scalable to e.g. using one million variants. However, when computing `cor(Gi x T, Gj x T)`, even if $\Sigma_{i,j} = 0$, `cor(Gi x T, Gj x T)` would be non-zero, therefore having to store and use a full LD matrix again. And LD blocks would not be valid anymore.
- The supplementary methods seem to suggest that LDpred2-auto is used here, while the code (https://github.com/cran/PRSPGx/blob/master/R/PRS_Dis_LDpred2.R) seems to suggest that fixed values for h^2 and p are used instead? Also a full dense correlation matrix is used?
- Are the PRS-PGx methods trained from ~ 3400 ($= 5661 * 3/5$) individuals in the real data analysis? I am very surprised you can get such good predictions from such a small sample size.
- How many variants are used in the real data analysis? 9,407,967 variants are mentioned in the methods, but as mentioned before, that is not possible given the expected poor scalability of the developed methods.

Minor comments:

- The name of the methods (PRS-PGx) is slightly unfortunate given that the main application is to a continuous trait (not a disease).
- Maybe it could be made clearer that the penalized regression methods are actually individual-level data methods. How can these be worse than the C+T method?
- The code used at https://github.com/cran/PRSPGx/blob/master/R/PRS_PGx_Bayes.R#L69 seems to be assuming that `cor(Gi x T, Gj)` = `cor(Gi, Gj x T)`, which does not seem to be true.
- I don't know if it is very often the case in RCTs that $\mu_T = 0.5$, but if so, would it be interesting to also derive equations for this special case?
- I find it odd to denote a covariance (e.g. `cov(Gi, Gj)`) by σ^2 since it can have a negative value.
- To be sure I understand the code of PRS-PGx-CT, the clumping is performed on the `cor(Gi, Gj)` while the thresholding is performed on the p-values from the model with the interaction term, and then the same SNPs are used for keeping variants in both types of effects?

- Is it really useful to talk about the special case of $\mu_T = 0$?
- Could be reworded or corrected:
 - + "its genotype main and genotype-by-treatment interaction effects"
 - + "for the simplicity"
 - + "an probability"
 - + for writing expectations, maybe use "E[X]" instead of "EX"
 - + $G \sim \text{Binomial}(2; \text{MAF}) \rightarrow G \sim \text{Binomial}(2; \text{AF})$
 - + define "GEI"

Reviewer #2 (Remarks to the Author):

Zhai et al. propose a method to model PRS for PGx studies by simultaneously modeling both the prognostic and predictive effects. They perform simulation studies and show that their newly developed PRS-PGx methods generally outperform the disease PRS methods with PRS-PGx-Bayes being superior. Application of PRS-PGx methods to a large cardiovascular RCT (IMPROVE-IT) to predict treatment related LDL cholesterol showed noticeable improvements.

This study is well-written and provides good data in both simulations and applied to a cardiovascular RCT for improved PRS methods for drug response prediction in PGx studies. These methods are needed, as the authors noted, most PGx PRS studies to date have relied on standard disease PRS methods which relies on stringent assumptions that are not necessarily correct and are restricted in terms of prediction potential and can result in limited power in prediction. By considering a PGx PRS that not only is built from variants with prognostic effects but also predictive effects related to treatment, an optimized PRS can be built to study treatment interaction effects for PRS in PGx studies.

I have a few comments and suggestions:

1. There has been intense focus on extending disease PRS to multi-ethnic populations. Studies have shown that PRS trained on European GWAS data have limited portability and reduced prediction in Non-European populations. The issue of multi-ethnic populations is absent in this study - how do these PGx PRS studies function with respect to trans-ancestry populations?
2. Although I understand this is primarily a methods paper demonstrating better prediction performance of PGx PRS methods for PGx studies, there is hardly any empirical results shown on the PGx PRS-Bayes method applied to the IMPROVE-IT study. Only r^2 and P-values are shown but it would be informative to show effect sizes and association results of PGx PRS-Bayes for the different arms of the IMPROVE-IT study.
3. Table 1: The five PRS methods have R^2 between 0.15 - 0.277, with the best performing being PRS-PGx-Bayes especially for two arms and T arm, which is what we expect for this method. However, the R^2 values are fairly low across all five PRS methods. Can the authors provide possible reasons for the low R^2 ? Does this indicate that PRS for PGx studies have low prediction in generally currently?
4. Following on comment above, for disease PRS, studies have reported the top X percentile of PRS vs. rest of distribution and highlight X fold odds ratio which can be informative for risk stratification. Can a similar logic be used here?

Minor edits:

5. Avoid superfluous words which appear to overhype the message of the study. For example "paradigm shift" in the Abstract could be toned down.

We would like to thank two reviewers for their critical review of our PRS-PGx manuscript. We appreciate their comments and questions, which led to a substantial improvement of the paper. Detailed responses to reviewers are given below. We highlighted the changes related to the comments in the revision.

Response to Reviewer 1

Zhai and colleagues present methods that use GWAS summary statistics and individual-level data from pharmacogenomics settings to build polygenic scores (PGS) with both good prognostic and predictive values. They compare their method with traditional PGS methods that use GWAS summary statistics only. I am not an expert in RCT, but I imagine this could have a major impact as a better use of PGS in RCTs. However, I have several comments and questions I would like the authors to address before I could recommend this for publication in Nature Communications. Overall, I fear that the size of the data used and the implementation of the new methods are not convincing enough.

Response: Thank you very much for the high-level summary of our research work in this manuscript. Please find below the detailed point-by-point response to each of the comments and we hope that we have sufficiently addressed the major concern that “the size of the data used and the implementation of the new methods are not convincing enough”.

Major comments

1 I am not sure I understand what are the prognostic effects (betas) in the model used; when we adjust for the baseline level Y_0 in the model, what would be the effects of the genetics on this change (i.e. $Y_1 - Y_0$ or Y_1/Y_0)?

Response: Thank you for your questions. As shown in the “Results” section, equation (1) $Y = X\gamma + \beta_T T + G\beta + (G \times T)\alpha + \epsilon$, β is a $m \times 1$ vector of prognostic effects (i.e., genotype main effects), α is a $m \times 1$ vector of predictive effects (i.e., genotype by treatment interaction effects). The prognostic effects β represent the association strength between the genotype and the drug response Y regardless of the treatment T . In contrast, the predictive effects α represent the additional association strength between the genotype and the drug response in the treatment arm (since we coded treatment arm as 1 and control arm as 0 in the model).

When we adjust for the baseline level Y_0 in the model, the genetic effect (β or α) on the change should be interpreted as the effects in addition to the baseline’s (Y_0 ’s) effect on the change. One of the major reasons that we adjust for Y_0 in the model is that the type I error of the baseline unadjusted model could be inflated for those variants associated with Y_0 , since the Y_0 is typically highly correlated with the change. In other words, the type I error inflation happens when baseline value Y_0 becomes a mediator between the variants and the change from baseline. This is the main conclusion from our another manuscript “A statistical perspective on baseline adjustment in pharmacogenomic genome-wide association studies of quantitative change” which has been recently accepted by *npj Genomic Medicine*.

References:

- [1] Zhang, H., Chhibber, A., Shaw, P.M., Mehrotra, D.V. & Shen, J. A statistical perspective

on baseline adjustment in pharmacogenomic genome-wide association studies of quantitative change. *npj Genomic Medicine* (2022). Accepted.

2 PRS-PGx-Bayes seems to be somewhat an extension of the PRS-CS method (using very similar Bayesian setting and tricks like regularizing $\phi_i \times \psi_i \leq 1$ that are not really mentioned as borrowed from there); can this really be considered a novel method?

Response: Thanks for the question. We agree with the reviewer that our proposed PRS-PGx-Bayes method shared the same Bayesian regression framework with the PRS-CS method. We have already referred to Ge et al. (2019) when describing the details of the methods in the “Methods: PRS-PGx-Bayes” section. To further avoid potential confusion, we revised the following sentences in the “Introduction” section:

“... Moreover, by extending the idea of global-local scaling parameters from disease GWAS (Ge et al., 2019) to PGx GWAS, the PRS-PGx-Bayes method is able to infer the posterior prognostic and predictive effects simultaneously ...”

In the “Methods: PRS-PGx-Bayes” section, we revised the following sentences:

“... Also as proposed by Ge et al. (2019), to avoid numerical issues caused by collinearity between SNPs, we set $\phi\psi_j \leq \rho$ and $\phi\xi_j \leq \rho$...”

Although the proposed PRS-PGx-Bayes method could be considered as an extension of the PRS-CS method from disease PRS space to PGx PRS space, the PRS-PGx-Bayes method we proposed has several unique features. Below we briefly summarize the novelties of the PRS-PGx-Bayes method in terms of the concept, the model, and the technicality.

Concept. The PRS-PGx-Bayes method (and other PRS-PGx methods) constructs both prognostic score and predictive score by incorporating the genetic main effect and the gene-by-treatment effect simultaneously to improve drug response prediction. To our best knowledge, no existing methods including the PRS-CS method can be directly applied to jointly model both prognostic and predictive effects for drug response prediction. The necessity of PGx-PRS approaches is validated by the proof of extremely stringent assumptions needed for the PRS-Dis approach to predict drug response. We believe that our theoretical framework and the concept of jointly modelling both prognostic and predictive effects for the construction of PGx PRS in PGx studies are novel in the field.

Model. To implement the concept of joint modelling of main and interaction effects, we extended the Bayesian model used in the PRS-CS method to accommodate both effects and their correlation in the PRS-PGx-Bayes method. Therefore, a model of scalar β becomes a model of vector (β, α) . A single variance parameter of β becomes a variance-covariance matrix of (β, α) . A joint distribution of prognostic and predictive effects is needed when considering the posterior instead of a univariate distribution. We believe that this systematic extension provides more appropriate methods and tools for polygenic score application in PGx studies.

Technicality. In terms of the implementation details, compared with the PRS-CS method, the PRS-PGx-Bayes method proposed to use the hierarchical half-t prior (Huang & Wand,

2013) to model the variance-covariance matrix of a multivariate normal distribution. Although the inverse Wishart (IW) prior may seem to be a more natural extension of Ge et al. (2019) and a more common practice when modelling covariance matrix, the IW prior imposes a dependency between the correlations and the variances: larger variances automatically imply the absolute value of the correlation near one while small variances imply the correlation near zero (Tokuda et al., 2011; Alvarez et al., 2014). We believe that the adoption of the hierarchical half-t prior in our PRS-PGx-Bayes method provides a better approach for modeling the variance-covariance matrix of a multivariate normal distribution in PGx applications.

References:

- [1] Ge, T., Chen, C.-Y., Ni, Y., Feng, Y.-C. A. & Smoller, J. W. Polygenic prediction via Bayesian regression and continuous shrinkage priors. *Nature Communications* **10**, 1776 (2019).
- [2] Huang, A. & Wand, M. P. Simple marginally noninformative prior distributions for covariance matrices. *Bayesian Analysis* **8**, 439–452 (2013).
- [3] Tokuda, T., Goodrich, B., Van Mechelen, I., Gelman, A. & Tuerlinckx, F. Visualizing distributions of covariance matrices. *Technical Report, Department of Psychology, University of Leuven* (2011).
- [4] Alvarez, I., Niemi, J. & Simpson, M. Bayesian inference for a covariance matrix. *Preprint at <https://arxiv.org/abs/1408.4050>* (2014).

3 Implementations of all methods presented here rely on very crude R code, e.g. calling `cor()` on the whole genotype matrix (i.e. computing and storing the full dense LD matrix); this seems highly inefficient and would never scale to the current data sizes used in summary statistics (e.g. even when restricting to e.g. 1M HapMap3 variants).

Response: Thank you for the comment. To avoid confusion, we would like to point out that our algorithm including `cor()` was applied on LD blocks (Berisa & Pickrell, 2016) in both simulations and real data analysis by recognizing that it is impossible to call `cor()` on the whole genome or chromosomes. The LD blocks were defined in Berisa & Pickrell (2016) and were publicly available at <https://bitbucket.org/nygcresearch/lddetect-data>. The same “by LD block” strategy was also adopted by Ge et al. (2019). More detailed justifications about this strategy are given in our response to Comment # 4.

To directly address reviewer’s concern on the computational costs of `cor()`, we conducted additional simulations by applying `cor()` on the largest LD block (chr 6, LD block 33). The simulation results were presented in Response Figure 1. After matching IMPROVE-IT PGx data to 1000 Genomes (1KG) data, we had 11,769 SNPs left in this block. We then evaluated the computational time and memory by randomly choosing 1,000, 3,000, 5,000, 7,000, 9,000 and the full 11,769 SNPs from the block. As expected, the computational time and memory increased at the rate of m^2 , where m denotes the number of variants. But even for the largest LD block, calling `cor()` took around 3 minutes and 1GB memory, which is quite reasonable.

Response Figure 1: Computational costs of calling `cor()` on chromosome 6 block 33 in terms of (a) computational time, (b) computational memory. Number of variants = 1,000, 3,000, 5,000, 7,000, 9,000, and 11,769 (whole LD block).

To further clarify the computational burden, in the Supplemental materials, we added the distribution of block sizes (i.e., the number of SNPs in each LD block) as the Supplementary Figure 12. In European population, there are 1,725 independent LD blocks across the whole genome. The largest LD block in the IMPROVE-IT PGx data is on chromosome 6 with 18,279 SNPs; after matching to 1KG, 11,769 SNPs are left to be handled simultaneously.

In the “Methods: PRS-PGx-Bayes” section, we revised the following paragraph:

“... In addition, we partition the genome into 1,725 largely independent genomic regions (Berisa & Pickrell, 2016) estimated using data from the 1000 Genomes European sample, and further conduct multivariate update of the effect sizes within each LD block. The distributions of block sizes in 1KG and IMPROVE-IT data are summarized in Supplementary Figure 12 a and b, respectively. The largest LD block (chr 6, block 33) after matching IMPROVE-IT PGx data to 1KG contains 11,769 SNPs in total ...”

Supplementary Figure 12: Distribution of LD block sizes where SNPs are included in (a) 1KG, and (b) IMPROVE-IT. The total number of independent LD blocks is 1,725 for European population.

We also conducted additional simulation to evaluate the total computational time of PRS-PGx-Bayes function using the largest LD block (i.e., chr 6, LD block 33). The results were presented in Supplementary Figure 9. The total computational time were assessed using PRS-PGx-Bayes function with 1,000 MCMC iterations, which also increased at the rate of m^2 to m^3 , where m denotes the number of variants. It took 43.5 hours for the largest LD block and 3.4 hours for the median LD block. It is worth mentioning that the computation in each LD block is independent, meaning we can easily distribute the computation of 1,725 LD blocks under the High Performance Computing environment to reduce the run time. In the authors’ working environment, the IMPROVE-IT whole genome analysis took about 40 hours, where typically 200 - 400 jobs were run simultaneously.

Supplementary Figure 9: Computational time on the largest LD block (chr 6, block 33) by running PRS-PGx-Bayes function with 1,000 MCMC iterations. Number of variants = 1,000, 3,000, 5,000, 7,000, 9,000, and 11,769 (whole LD block). The real genetic data was obtained from the IMPROVE-IT trial with the sample size of 5,661. The effect sizes and phenotype data were simulated with heritability fixed at 0.3, $\psi/\xi = 1$, and $P(\text{causal}) = 0.01$.

In the “Results: Simulation studies” section, we added a subsection of “Computational time”:

“... To assess the computational burden of the proposed method, we applied the PRS-PGx-Bayes function with 1,000 MCMC iterations to chromosome 6, LD block 33 (the largest LD block with 11,769 SNPs). As a sensitivity analysis, we also explored scenarios by randomly choosing 1,000, 3,000, 5,000, 7,000, 9,000 SNPs from that block. The real genetic data was obtained from the IMPROVE-IT trial with a sample size of 5,661. The effect sizes and phenotype data were simulated with heritability fixed at 0.3, $\psi/\xi = 1$, and $P(\text{causal}) = 0.01$. The tuning parameters were selected via cross-validation. The computation was completed on a single core of 2.4 GHz Intel Core i5. We summarized the result in Supplementary Figure 9, which shows that the computational time increased at the rate of m^2 to m^3 , where m denotes the number of variants. The result also shows that it took roughly 43.5 hours for the largest LD block and 3.4 hours for the median-size LD block (Supplementary Figure 12) to complete the computation. In practice, since the computation in each LD block is independent, we could further shorten the computational time by parallel computing 1,725 LD blocks across the whole genome. In the authors’ High Performance Computing working

environment, the IMPROVE-IT whole genome analysis took about 40 hours, where typically 200 - 400 jobs were run simultaneously. ...”

References:

- [1] Berisa, T. & Pickrell, J. K. Approximately independent linkage disequilibrium blocks in human populations. *Bioinformatics* **32**, 283 (2016).
- [2] Ge, T., Chen, C.-Y., Ni, Y., Feng, Y.-C. A. & Smoller, J. W. Polygenic prediction via Bayesian regression and continuous shrinkage priors. *Nature Communications* **10**, 1776 (2019).

4 On the same note, the correlation/LD matrix between variants can often be considered as sparse (banded or even block-diagonal). This is one of the main reasons PGS methods can be scalable to e.g. using one million variants. However, when computing $\text{cor}(G_i \times T, G_j \times T)$, even if $\text{Sigma}_{i,j} = 0$, $\text{cor}(G_i \times T, G_j \times T)$ would be non-zero, therefore having to store and use a full LD matrix again. And LD blocks would not be valid anymore.

Response: Thanks for the insightful comment. We agree with the reviewer that even if SNP i and SNP j are independent, $\text{cor}(G_i \times T, G_j \times T)$ would still be non-zero. In fact, we can prove that under typical 1:1 treatment-placebo allocation ratio in randomized clinical trials (please see the our proof added to Supplementary Method D; also see response to Comment # 11 for details),

$$\text{cor}(G_i \times T, G_j \times T) = \sqrt{f_i f_j},$$

where f_i, f_j are the MAFs of SNP i and j , respectively. When the treatment-placebo allocation ratio is not 1:1, in addition to MAFs, only the percentage of treatment patients (p) is needed to determine $\text{cor}(G_i \times T, G_j \times T)$. Therefore, if needed, we don't have to store a full LD matrix to track values of $\text{cor}(G_i \times T, G_j \times T)$. Instead, we only need to record a vector of f_i 's and a scalar p .

On the validity of the LD blocks, we want to point out that the “by LD block” implementation strategy was the choice we made to balance the computational costs and the prediction accuracy. We believe that the accuracy loss from ignoring $\text{cor}(G_i \times T, G_j \times T)$ is acceptable in exchange for feasible computation time in the whole genome data analysis.

To validate this point, we performed additional simulation studies using block 31 and block 32 on chromosome 19, following the same simulation settings in the main text. The PRS-PGx-Bayes function was applied on these blocks separately (i.e., block-by-block) and jointly (i.e., full LD matrix was used). We also applied PRS-PGx-Bayes method to the uniform blocks with size = 200, 500, and 2,500 (i.e., the number of variants in each block = 200, 500, and 2,500). Similarly, as Fig. 2 in the main text, the heritability was fixed at 0.3 and $\psi/\xi = 1$. The numbers of the causal variants for $P(\text{causal}) = 0.001, 0.01$ and 0.1 were 5, 50 and 500, respectively. The training sample size was 3,000. The tuning parameters were selected via cross-validation in the training data. Performance comparisons of PRS-PGx-Bayes method with different implementation strategies were summarized and added to Supplementary Figure 4.

Overall, there is indeed a slightly decreasing trend in R^2 and $-\log(\text{p-value})$ from using the full genotype matrix to using uniform blocks with size 200. However, such differences across different

types of blocks are limited, especially between LD blocks and full genotype matrix. For example, when $p(\text{causal}) = 0.001$, compared to full genotype matrix (orange bar), the LD block approach (red bar) decreases R^2 by 0.4%. We also summarized the relative decreases of LD block approach compared to the full genotype approach in Supplementary Table 1. The table shows that the relative R^2 decreases of the LD block approach in simulation studies are very little (i.e., all $\leq 1.1\%$).

Supplementary Figure 4: Performance comparisons of PRS-PGx-Bayes method when carried out by different implementation strategies (based on LD blocks 31 and 32 on chromosome 19) in simulation studies, where heritability was fixed at 0.3, training sample size = 3,000, and $\psi/\xi = 1$. PRS-PGx-Bayes was carried out by the uniform block with size 200, 500, and 2,500, the LD block, and the full genotype data. The numbers of the causal variants for $P(\text{causal}) = 0.001, 0.01$ and 0.1 were 5, 50 and 500, respectively. The tuning parameters of PRS-PGx-Bayes were selected via cross-validation in the training data. The error bar represents the standard error of the results calculated from the testing sets. The performance was assessed in the testing set in terms of (a) prediction accuracy R^2 of S_{PGx} in two arms, (b) p-value for the predictive effect of $S_{pred} \times T$ interaction, (c) R^2 of S_{PGx} under treatment arm, and (d) R^2 of S_{PGx} under control arm.

Supplementary Table 1: Performance decreases of PRS-PGx-Bayes method carried out from using full genotype matrix (i.e., using LD blocks 31 and 32 jointly on chromosome 19) to using LD blocks (i.e., using LD blocks 31 and 32 separately on chromosome 19) in simulation studies. Heritability was fixed at 0.3, training sample size = 3,000, and $\psi/\xi = 1$. The numbers of the causal variants for $P(\text{causal}) = 0.001, 0.01$ and 0.1 were 5, 50 and 500, respectively. The performance was assessed in terms of prediction accuracy R^2 of S_{PGx} in two arms, p-value for the predictive effect of $S_{pred} \times T$ interaction, R^2 of S_{PGx} under treatment arm, and R^2 of S_{PGx} under control arm.

P(causal)	0.001	0.01	0.1
R^2	0.4%	0.7%	0.9%
Pred P-value	0.6%	0.6%	0.9%
R^2 : T arm	0.3%	0.5%	0.8%
R^2 : C arm	0.5%	0.7%	1.1%

To summarize what we just discussed, in the “Results: Simulation studies” section, we added the following paragraph:

“... To further assess the impact of different implementation strategies, we also performed additional simulations where PRS-PGx-Bayes function was applied on LD blocks jointly (i.e., full LD matrix across LD blocks was used). The simulation settings remained the same as described in Fig. 2. As a sensitivity analysis, we also applied PRs-PGx-Bayes method to the uniform blocks with number of variants in each block as 200, 500, and 2,500, respectively. Supplementary Figure 4 shows that there is a slightly decreasing trend in R^2 and $-\log(\text{p-value})$ from using the full genotype matrix to using uniform blocks with size 200. However, such differences across different types of blocks are limited, especially between LD blocks and full genotype matrix. For example, when $P(\text{causal}) = 0.001$, compared to using the full genotype matrix, the LD block approach only decreases R^2 by 0.4%. The relative decreases of LD block approach compared to the full genotype approach was summarized in Supplementary Table 1. The table shows that the relative decreases of the LD block approach in simulation studies are very small (i.e., all $\leq 1.1\%$) ...”

In the “Methods: PRS-PGx-Bayes” section, we added the following sentences:

“... As shown in Supplementary Figure 4 and Supplementary Table 1, this strategy is justified by that fact that only a small relative difference ($\leq 1.1\%$) is observed when the PRS-PGx-Bayes function is carried out by LD blocks, compared to across multiple LD blocks in simulation studies ...”

In the “Discussion” section, we added the following sentences:

“... Lastly, we suggest applying our proposed methods by LD blocks. Expanding the size of blocks (e.g., using the whole chromosome) may slightly improve prediction accuracy but also significantly increase computational costs. On the other hand, further reducing the size of blocks (e.g., using the uniform blocks with smaller size) can reduce run-time but also possibly increase the bias by missing long-range LD. We believe that the by LD block strategy is a right trade-off between decent

modeling accuracy and feasible computational time. We also acknowledge that future work is needed to further improve the computational efficiency to incorporate more SNPs for simultaneous analysis ...”

References:

- [1] Mak, T. S. H., Porsch, R. M., Choi, S. W., Zhou, X. & Sham, P. C. Polygenic scores via penalized regression on summary statistics. *Genetic Epidemiology* **41**, 469–480 (2017).

5 *The supplementary methods seem to suggest that LDpred2-auto is used here, while the code (https://github.com/cran/PRSPGx/blob/master/R/PRS_Dis_LDpred2.R) seems to suggest fixed values for h^2 and p are used instead? Also a full dense correlation matrix is used?*

Response: Thanks for pointing out this inconsistency. We used LDpred2-grid method (Privé et al., 2020) for the implementation of PRS-Dis-LDpred2 in our manuscript. Per your suggestion, we changed the description of PRS-Dis-LDpred2 in Supplementary Method C to the following:

“... As an updated version of LDpred, LDpred2-grid (Privé et al., 2020) introduces a third hyper-parameter indicating whether sparsity is enabled or not. When implementing the PRS-Dis-LDpred2 method, we tested a grid of hyper-parameters with p and h^2 . We also enabled the sparsity by setting “sparse = TRUE”, aimed at providing sparse effect size estimates, i.e., shrinking some effects to exactly 0 ...”

The reason why we used LDpred2-grid, instead of LDpred2-auto, for our simulation studies and real data analysis was because although their performances were generally comparable, we found that LDpred2-grid slightly outperformed LDpred2-auto in many scenarios (e.g., the proportion of causal variants was small to moderate) in our additional simulations.

Specifically, as presented in the Supplementary Figure 14, we conducted additional simulation studies following the same settings described in the “Methods: Simulation” section. The heritability was fixed at 0.3 and $\psi/\xi = 1$. Numbers of the causal variants for $P(\text{causal}) = 0.001, 0.01$ and 0.1 were 5, 50 and 500, respectively. The training sample size was 20,000. The tuning parameters of LDpred2-grid were selected via cross-validation in the training data. We compared LDpred2-grid with LDpred2-auto using the same performance metrics described in Fig. 2, which were R^2 of S_{PGx} in the whole population, treatment arm, and control arm, respectively, and the predictive p-value of $S_{pred} \times T$ interaction. As we could see, when $P(\text{causal})$ was low or moderate (i.e., $P(\text{causal}) = 0.001$ or 0.01), LDpred2(-grid) was slightly better than LDpred2-auto; and when $P(\text{causal})$ was high (i.e., $P(\text{causal}) = 0.1$), LDpred2-auto outperformed LDpred2(-grid) a little bit. However, overall the two methods’ performance was very close to each other. These observations were consistent with Privé et al. (2020).

We added Supplementary Figure 14 to the Supplemental materials document.

Supplementary Figure 14: Predictive performance of LDpred2-grid and LDpred2-auto methods in the simulation studies, where heritability was fixed at 0.3 and $\psi/\xi = 1$. Numbers of the causal variants for $P(\text{causal}) = 0.001, 0.01$ and 0.1 were 5, 50 and 500, respectively. The training sample size was 20,000. The tuning parameters of LDpred2-grid were selected via cross-validation in the training data. The error bar represents the standard error of the results calculated from the testing sets. The performance was assessed in the testing set in terms of (a) prediction accuracy R^2 of S_{PGx} in two arms, (b) p-value for the predictive effect of $S_{pred} \times T$ interaction, (c) R^2 of S_{PGx} under treatment arm, and (d) R^2 of S_{PGx} under control arm.

In Supplementary Method C, we added the following paragraph:

“... Besides LDpred2-grid, Privé et al. (2020) also proposed another version called LDpred2-auto, which estimates p and h^2 within the model. This makes LDpred2-auto a method free of hyper-parameters. More specifically, to estimate p in the Gibbs sampler, LDpred2 counts the number of non-zero variants as $M_c = \sum_j (\beta_j \neq 0) \sim \text{Binomial}(m, p)$; and estimates $h^2 = \beta^T \mathbf{R} \beta$, where \mathbf{R} is the correlation matrix. To determine which method, LDpred2-grid or LDpred2-auto, should be used as PRS-Dis-LDpred2 function, we further conducted simulation studies following the same simulation method described in the main text. The heritability was fixed at 0.3 and $\psi/\xi = 1$. Numbers of the causal variants for $P(\text{causal}) = 0.001, 0.01$ and 0.1 were 5, 50 and 500, respectively. The training sample size was 20,000. The tuning parameters of LDpred2-grid were selected via cross-validation in the training data. Supplementary Figure 14 indicated that although overall the two methods’ performance was very close to each other, LDpred2-grid actually slightly outperformed LDpred2-auto in many scenarios (i.e., when the proportion of causal variants was small to moderate). Therefore, in our study, we only used LDpred2-grid as the best disease PRS method to be compared with our proposed PRS-PGx approaches ...”

The PRS-Dis-LDpred2 function (a disease PRS approach) was also applied on LD blocks simultaneously (assuming independence among different LD blocks). To further avoid potential confusion, in Supplementary Method C, we added the following sentences:

“... We directly use the disease PRS LDpred2 method by LD blocks as the PRS-Dis-LDpred2 method ...”

References:

- [1] Privé, F., Arbel, J. & Vilhjálmsson, B. J. LDpred2: better, faster, stronger. *Bioinformatics* **36**, 5424–5431 (2020).

6 Are the PRS-PGx methods trained from ~ 3400 ($=5661 * 3/5$) individuals in the real data analysis? I am very surprised you can get such good predictions from such a small sample size.

Response: Thank you for your question and comment. In our analysis of the IMPROVE-IT GWAS data, the PRS-PGx methods were trained from $\sim 4,528$ ($= 5,661*4/5$) individuals based on a 5-fold cross-validation procedure (i.e., the PGx GWAS summary statistics were obtained based on the training set of $\sim 4,528$ individuals). The information about the CV procedure was provided in the “Methods: IMPROVE-IT PGx GWAS data analysis” section.

Regarding the concern about getting good predictions from a small sample, we would like to point out that 1) a sample size of 5,661 might be small in disease GWAS but is actually considered large in PGx studies since the patients are from clinical trials; 2) the good prediction results probably came from the fact that drug responses typically have larger genetic effects and higher heritability. Hence, smaller (compared to disease GWAS) sample size is adequate for good prediction in PGx GWAS setting.

For example, Harper and Topol (2012) examined all of the GWAS catalogued in the US National Human Genome Research Institute at the time and showed that pharmacogenomics studies are sevenfold more likely to achieve odds-ratio (OR) > 3.0 compared to common disease GWAS in which the ORs are typically in the range of 1.05 to 1.15. Maranville and Cox (2016) also “found significantly larger effect sizes for studies focused on pharmacogenomic phenotypes, as compared to complex disease risk, morphological phenotypes, and endophenotypes.”

In summary, although sample sizes from PGx studies are usually smaller than those in disease GWAS, the significantly larger effect sizes from PGx studies can result in decent power to detect variants associated with drug response phenotypes and further enable good prediction performance of PRS for drug response prediction in PGx studies.

References:

- [1] Harper, A. R., & Topol, E. J. Pharmacogenomics in clinical practice and drug development. *Nature Biotechnology* **30**, 1117–1124 (2012).
- [2] Maranville, J. C., & Cox, N. J. Pharmacogenomic variants have larger effect sizes than genetic

variants associated with other dichotomous complex traits. *The Pharmacogenomics Journal* **16**, 388–392 (2016).

7 How many variants are used in the real data analysis? 9,407,967 variants are mentioned in the methods, but as mentioned before, that is not possible given the expected poor scalability of the developed methods.

Response: Thanks for your question. Yes, 9,407,967 variants were used in the real data analysis. As we discussed in response to Comment # 3, our functions were applied by LD blocks (instead of by chromosome or by whole genome). The block sizes of IMPROVE-IT data range from 126 to 18,279. Please refer to the response to Comment # 3 for the block size distribution and the related computational time illustration. In summary, we believe our implementation strategy is a right trade-off between maintaining decent modeling accuracy and reducing computational costs. The total computation time for the whole genome data analysis using the PRS-PGx methods is reasonable (in the authors' working environment, the IMPROVE-IT whole genome analysis carried out by PRS-PGx-Bayes algorithm took about 40 hours, where typically 200 - 400 jobs were run simultaneously).

Minor comments

8 The name of the methods (PRS-PGx) is slightly unfortunate given that the main application is to a continuous trait (not a disease).

Response: Thank you for your comment. We agree with the reviewer that Polygenic Risk Score (PRS) sounds more like a score for disease risk prediction, as opposed to a drug response prediction. We chose the name PRS to be consistent with the terminology used in the PRSice method and the PRS-CS method.

We would also like to point out that, although we are doing drug response prediction, it can be interpreted as the risk prediction of not having desirable outcomes, such as not responding to the treatment or elevated incidence rates of adverse events. Similar to disease phenotypes, there are usually three types of drug response phenotypes in pharmacogenomics studies: continuous (e.g., LDL change from baseline), binary (e.g., adverse drug reaction) and time-to-event (e.g., the time to cardiovascular death).

Although we mainly implement the PRS-PGx methods in the analysis of continuous drug response phenotypes, all of the PRS-PGx methods, except for PRS-PGx-Bayes, can be directly applied to binary and survival phenotypes. Furthermore, it is possible in future research to extend the PRS-PGx-Bayes to analyzing binary and survival endpoints by adopting the Bayesian logistic regression and Bayesian Cox proportional hazards model instead of the Bayesian linear regression as mentioned in the "Discussion" section.

9 Maybe it could be made clearer that the penalized regression methods are actually individual-level data methods. How can these be worse than the C+T method?

Response: Thanks for the suggestion. We agree to make it clearer that the penalized regression methods use individual-level data. We revised the following sentences in the "Introduction" section:

“... These methods include PRS-PGx-Unadj (Unadjusted), PRS-PGx-CT (Clumping + Thresholding), PRS-PGx-L, -GL, -SGL (-Lasso, -Group Lasso, -Sparse Group Lasso regression) methods. Our proposed methods use only PGx genome-wide association summary statistics and an external LD reference panel except for the penalized regression based methods, which require access to individual level genetic and phenotypic data. ...”

In the “Methods: PRS-PGx-L, -GL and -SGL” section, we also added the following sentences:

“... In the PRS-PGx-L, -GL and -SGL methods, penalized regression is applied to estimate equation (1) with individual-level data ...”

Regarding the question about the performance comparison between the penalized regression methods and the C+T method, we would like to clarify that, in general, two factors may impact the comparison: 1) the difference in the data that was used; 2) the difference in the methodology.

On the first point, using individual-level data may not necessarily increase the prediction performance in a significant way. Lin and Zeng (2010a, 2010b) showed that using GWAS summary statistics is not worse than the individual-level data as long as both analyses are conducted under the same model and assumptions. To quote directly from Lin and Zeng (2010b), “We show theoretically and numerically that meta-analysis of summary results is statistically as efficient as joint analysis of individual participant data”.

On the second point, we would like to point out that although Lasso based methods may be more sophisticated, their performances are not guaranteed to be higher than the simpler methods like C+T. For example, in the paper of Lassosum (Mak et al., 2017), their Figure 2B (the real data analysis results) also shows that for some particular diseases, i.e., Bipolar disorder (BD) and Type 2 diabetes (T2D), C+T could outperform the Lasso based method. We added Figure 2B of Mak et al., (2017) here as the Response Figure 2. In our analysis, there are scenarios where Lasso based methods outperformed the method C+T. For example, Supplementary Figure 5 shows that when the heritability was 0.5, PRS-PGx-GL was much better than PRS-PGx-CT in terms of both prediction accuracy and patient stratification. In Supplementary Figure 7, when the proportional of causal variants was low (i.e., the effect size of each causal variant was large), PRS-PGx-SGL was better than, or at least comparable to, PRS-PGx-CT.

In summary, we believe the C+T and Lasso based methods have their own advantages and disadvantages depending on the heritability and the number of causal variants. Therefore, the performance comparison between C+T and Lasso is ultimately trait-specific in disease PRS; and similarly, the comparison is drug-response-specific in PGx PRS.

In the “Discussion” section, we added the following sentences:

“... Interestingly, we find that the C+T method (PRS-PGx-CT) can outperform the penalized regression based methods (PRS-PGx-L, PRS-PGx-GL and PRS-PGx-SGL) in some of the scenarios. This pattern was also observed in Mak et al., (2017). Our results suggest that Lasso based methods are sensitive to the noise, and perform poorly when the signal-to-noise ratio is small (Supplementary Figure 5 and 7). Further study is needed to examine the difference more comprehensively ...”

References:

[redacted]

Response Figure 2: Performance of Lassosum vs. other methods when using real summary statistics data from meta-analyses. Predictive accuracy was assessed by prediction in the WTCCC dataset after the contribution from WTCCC was removed from the summary statistics. (Mak et al., 2017) [Editorial note: this figure was redacted due to third-party rights. It can be found in Mak et al., 2017 [3], Figure 2B]

- [1] Lin, D. Y. & Zeng, D. On the relative efficiency of using summary statistics versus individual-level data in meta-analysis. *Biometrika* **97**, 321–332 (2010a).
- [2] Lin, D. Y. & Zeng, D. Meta-Analysis of Genome-Wide Association Studies: No Efficiency Gain in Using Individual Participant Data. *Genetic Epidemiology* **34**, 60–66 (2010b).
- [3] Mak, T. S. H., Porsch, R. M., Choi, S. W., Zhou, X. & Sham, P. C. Polygenic scores via penalized regression on summary statistics. *Genetic Epidemiology* **41**, 469–480 (2017).

10 The code used at https://github.com/cran/PRSPGx/blob/master/R/PRS_PGx_Bayes.R L69 seems to be assuming that $\text{cor}(G_i \times T, G_j) = \text{cor}(G_i, G_j \times T)$, which does not seem to be true.

Response: Thanks for pointing this out. We agree with you that in general $\text{cor}(G_i \times T, G_j) \neq \text{cor}(G_i, G_j \times T)$. However, we want to clarify that we treat these two terms approximately equal because we can show in theory (added to the Supplementary Method D) that the difference is pretty small and has little impact on the performance of the method we proposed. We used this approximation under the belief that it can help reduce the computational cost. However, as shown in the Response Table 1, the reduction is limited. We will revise our PRS-PGx-Bayes function in the PRSPGx R package in the next CRAN version update.

To evaluate the difference between these two terms, we make the following assumptions. Let f_i be the MAF of G_i . Under Hardy-Weinberg equilibrium, $\mu_i = E[G_i] = 2f_i$, $\sigma_i^2 = \text{var}(G_i) = 2f_i(1 - f_i)$.

We use subscript j and T for G_j and T , respectively. σ_{ij} denotes the covariance between G_i and G_j . Further assume T and G_i (or G_j) are independent, we derived (in Supplementary Method D) that:

$$\begin{aligned} \text{cor}(G_i \times T, G_j) - \text{cor}(G_j \times T, G_i) &= \frac{\sigma_{ij}\mu_T}{\sqrt{\text{var}(G_i T)\text{var}(G_j)}} - \frac{\sigma_{ij}\mu_T}{\sqrt{\text{var}(G_j T)\text{var}(G_i)}} \\ (\text{let } \mu_T = 1/2) &= \frac{\sigma_{ij}}{\sigma_j\sqrt{\sigma_i^2 + \sigma_i^2 + \mu_i^2}} - \frac{\sigma_{ij}}{\sigma_i\sqrt{\sigma_j^2 + \sigma_j^2 + \mu_j^2}} \\ &= \text{cor}(G_i, G_j) \left(\frac{1}{\sqrt{2 + \mu_i^2/\sigma_i^2}} - \frac{1}{\sqrt{2 + \mu_j^2/\sigma_j^2}} \right) \\ &= \text{cor}(G_i, G_j)(\sqrt{1 - f_i} - \sqrt{1 - f_j})/\sqrt{2}. \end{aligned}$$

Without loss of generality, assume $0 < f_i \leq f_j \leq 0.5$. Follow the deduction in VanLiere and Rosenberg (2008), it can be shown that $-\sqrt{r_j}\sqrt{r_i} \leq \text{cor}(G_i, G_j) \leq \sqrt{r_i}/\sqrt{r_j}$, where $r_i = f_i/(1 - f_i)$, $r_j = f_j/(1 - f_j)$. By Lagrangian multipliers, we can show that

$$|\text{cor}(G_i \times T, G_j) - \text{cor}(G_j \times T, G_i)| \leq 0.06626,$$

where the maximum is attained at the boundary $f_j = 0.5$, $f_i = 1 - \sqrt[3]{0.5} \approx 0.2063$ and $\text{cor}(G_i, G_j) = \pm\sqrt{\sqrt[3]{2} - 1} \approx \pm 0.5098$.

To further investigate how this approximation would affect the predictive performance, we re-ran the simulation following the ‘‘Methods: Simulations’’ section in the main text. Simulation settings remained the same as described in Fig. 2. Four criteria were used to compare the performance of the algorithm with vs without the approximation: R^2 under the whole population, treatment arm, and control arm, as well as the predictive p-value. The comparison results were added into the Supplemental materials document as Supplementary Figure 15. The figure shows that there is very little difference between the algorithm with the approximation and without the approximation.

To summarize what we just discussed, in Supplementary Method D, we added the following paragraph:

‘‘... In practice, we approximated $\text{cor}(G \times T, G) \approx \text{cor}(G, G \times T)$ in the top-right block and the bottom-left block of **D**, which can reduce the computational cost of PRS-PGx-Bayes function. We can prove that such approximation held with the maximal difference smaller than 0.066 ... To further evaluate the impact of such approximation on the final results, we compared predictive performances of PRS-PGx-Bayes with and without approximation in the simulation studies, where simulation settings remained the same as described in Fig. 2 in the main text (i.e., the heritability was fixed at 0.3, the training sample size was 3,000, and $\psi/\xi = 1$). Supplementary Figure 15 suggested that there was very little difference in terms of predictive performances when the approximation was used ...’’

We introduced this approximation strategy because we believed it can help reduce the computational cost. The Response Table 1 was generated following the same simulation setting described in the Comment #3. Specifically, we checked the computational time of chr 6, LD block 33 (the largest LD block with 11,769 SNPs) based on one single core. We also explored scenarios by randomly choosing

Supplementary Figure 15: Predictive performance of PRS-PGx-Bayes with or without approximation (i.e., $\text{cor}(G \times T, G) \approx \text{cor}(G, G \times T)$) in the simulation studies, where heritability was fixed at 0.3 and $\psi/\xi = 1$. The numbers of the causal variants for $P(\text{causal}) = 0.001, 0.01$ and 0.1 were 5, 50 and 500, respectively. The training sample size was 3,000. The error bar represents the standard error of the results calculated from the testing sets. The performance was assessed in the testing set in terms of (a) prediction accuracy R^2 of S_{PGx} in two arms, (b) p-value for the predictive effect of $S_{pred} \times T$ interaction, (c) R^2 of S_{PGx} under treatment arm, and (d) R^2 of S_{PGx} under control arm.

1,000, 3,000, 5,000, 7,000, 9,000 SNPs from that block. The computational time was assessed using PRS-PGx-Bayes function with 1,000 MCMC iterations, and the tuning parameters were selected via cross-validation. It turned out that such approximation actually cannot reduce the computational time significantly (only around 1.5% reduction).

Response Table 1: Computational time comparisons when running PRS-PGx-Bayes function with/without approximation.

Number of SNPs	1,000	3,000	5,000	7,000	9,000	11,769
Without Approx (hr)	2.02	3.44	8.09	15.47	26.37	44.16
With Approx (hr)	1.99	3.38	7.98	15.31	26.03	43.52

References:

- [1] VanLiere, J. M., & Rosenberg, N. A. Mathematical properties of the r^2 measure of linkage disequilibrium. *Theoretical Population Biology* **74**, 130–137 (2008).

11 I don't know if it is very often the case in RCTs that $\mu_T = 0.5$, but if so, would it be interesting to also derive equations for this special case?

Response: Thanks for your question. Yes, $\mu_T = 0.5$ corresponds to a 1-to-1 treatment-placebo allocation ratio, which is the most commonly used ratio in randomized clinical trials. It is also possible that μ_T is different from 0.5 such as when the allocation ratio is 2-to-1 or 3-to-1 etc.

We agree that it would be more informative to provide some equations for the special cases. Therefore, we added the following paragraphs to Supplementary Method D:

Now we consider several special cases of:

$$\begin{aligned}\text{cor}(G_i \times T, G_j) &= \frac{\mu_T \sigma_{ij}}{\sqrt{(\mu_T^2 \sigma_i^2 + 4f_i^2 \sigma_T^2 + \sigma_i^2 \sigma_T^2) \sigma_j^2}}, \\ \text{cor}(G_i \times T, G_j \times T) &= \frac{(\mu_T^2 + \sigma_T^2) \sigma_{ij} + 4\sigma_T^2 f_i f_j}{\sqrt{(\mu_T^2 \sigma_i^2 + 4f_i^2 \sigma_T^2 + \sigma_i^2 \sigma_T^2)(\mu_T^2 \sigma_j^2 + 4f_j^2 \sigma_T^2 + \sigma_j^2 \sigma_T^2)}}.\end{aligned}$$

1. SNP i and SNP j ($i \neq j$) are independent (i.e., $\sigma_{ij} = 0$)

$$\begin{aligned}\text{cor}(G_i \times T, G_j) &= 0, \quad i \neq j, \\ \text{cor}(G_i \times T, G_j \times T) &= \frac{4\sigma_T^2 f_i f_j}{\sqrt{(\mu_T^2 \sigma_i^2 + 4f_i^2 \sigma_T^2 + \sigma_i^2 \sigma_T^2)(\mu_T^2 \sigma_j^2 + 4f_j^2 \sigma_T^2 + \sigma_j^2 \sigma_T^2)}}, \quad i \neq j.\end{aligned}$$

2. SNP i and SNP j ($i \neq j$) are independent (i.e., $\sigma_{ij} = 0$); assume $T \sim \text{Binomial}(1, p)$ (i.e., $\mu_T = p$, $\sigma_T^2 = p(1-p)$), since $G_i \sim \text{Binomial}(2, f_i)$ and $G_j \sim \text{Binomial}(2, f_j)$ (i.e., $\sigma_i^2 = 2f_i(1-f_i)$, $\sigma_j^2 = 2f_j(1-f_j)$), we have

$$\begin{aligned}\text{cor}(G_i \times T, G_j) &= 0, \quad i \neq j, \\ \text{cor}(G_i \times T, G_j \times T) &= \frac{2(1-p)f_i f_j}{\sqrt{f_i f_j (f_i - 2pf_i + 1)(f_j - 2pf_j + 1)}}, \quad i \neq j.\end{aligned}$$

Further assume $p = 0.5$ (most typical randomized clinical trial):

$$\begin{aligned}\text{cor}(G_i \times T, G_j) &= 0, \quad i \neq j, \\ \text{cor}(G_i \times T, G_j \times T) &= \sqrt{f_i f_j}, \quad i \neq j.\end{aligned}$$

3. SNP i and SNP j ($i \neq j$) are independent (i.e., $\sigma_{ij} = 0$) and $T \equiv 1$.

$$\begin{aligned}\text{cor}(G_i \times T, G_j) &= 0, \quad i \neq j, \\ \text{cor}(G_i \times T, G_j \times T) &= 0, \quad i \neq j.\end{aligned}$$

12 I find it odd to denote a covariance (e.g. $\text{cov}(G_i, G_j)$) by σ_{ij}^2 since it can have a negative value.

Response: Thanks for pointing this out. We have replaced σ_{ij}^2 by σ_{ij} in Supplementary Method D accordingly.

13 To be sure I understand the code of PRS-PGx-CT, the clumping is performed on the $\text{cor}(G_i, G_j)$ while the thresholding is performed on the p -values from the model with the interaction term, and then the same SNPs are used for keeping variants in both types of effects?

Response: Thanks for the question. The clumping is performed on the $\text{cor}(G_i, G_j)$ while the thresholding is performed on the 2-df test (i.e., joint test of $G + G \times T$) p -values. And then the same SNPs are used for keeping variants in both types of effects.

14 Is it really useful to talk about the special case of $\mu_T = 0$?

Response: Thanks for your question. We acknowledge that the special case of $\mu_T = 0$ is too simplified. Therefore, we now provide the deduction of $E[\beta_j|\hat{\beta}_j]$ and $E[\alpha_j|\hat{\alpha}_j]$ for $\mu_T = 0.5$ in Supplementary Method E. We deleted the special case of $\mu_T = 0$ and updated the ‘‘Methods: PRS-PGx-Bayes’’ section as the following:

‘‘... Second, assume a one-to-one treatment-placebo allocation ratio ($\mu_T = 0.5$), unlinked genetic markers ($\sigma_{ij} \equiv 0$ for $i \neq j$) and $\rho_j \equiv 0$ (i.e., within each SNP, the two effect sizes are independent). We can derive the formulas explicitly for $E[\beta_j|\hat{\beta}_j]$ and $E[\alpha_j|\hat{\alpha}_j]$ (Supplementary Method E). Under the simplified scenario where all markers’ MAFs are small, $f_j \equiv f \rightarrow 0$, we can show that:

$$\begin{aligned}E[\beta_j|\hat{\beta}_j] &\approx \frac{\hat{\beta}_j - \frac{c}{s_j}\hat{\alpha}_j}{t_j - c^2/s_j}, \\ E[\alpha_j|\hat{\alpha}_j] &\approx \frac{\hat{\alpha}_j - \frac{c}{t_j}\hat{\beta}_j}{s_j - c^2/t_j},\end{aligned}$$

where $t_j = 1 + \phi^{-1}\psi_j^{-1}$, $s_j = 1 - f + \phi^{-1}\xi_j^{-1}$ and $c = \sqrt{(1 - f)/2}$. To understand the above equations, we can interpret $1/t_j = \frac{1}{1 + \phi^{-1}\psi_j^{-1}}$, $1/s_j = \frac{1}{1 - f + \phi^{-1}\xi_j^{-1}}$ as the shrinkage factors. Therefore, $\hat{\beta}_j/t_j$ and $\hat{\alpha}_j/s_j$ are the ‘shrunk’ effects: $t_j = s_j = 1$ indicates no shrinkage while $t_j = s_j \rightarrow \infty$ yields full shrinkage. The correlation c between G_j and $G_j \times T$ also contributes to the second part of the numerator because the bias induced by the positive correlation needs to be corrected. ...’’

15 Could be reworded or corrected:

- + ‘‘its genotype main and genotype-by-treatment interaction effects’’
- + ‘‘for the simplicity’’
- + ‘‘an probability’’

- + for writing expectations, maybe use “ $E[X]$ ” instead of “ EX ”
- + $G \sim \text{Binomial}(2; \text{MAF}) \rightarrow G \sim \text{Binomial}(2; \text{AF})$
- + define “ GEI ”

Response: Thanks for pointing these out.

In the “Introduction” section, we revised the following sentence:

“... have a constant ratio between its genotype main effect and genotype-by-treatment interaction effect ...”

In the “Results: Conceptual framework of the PRS-PGx methods” section, we revised the following sentence:

“... For ~~the~~ simplicity ...”

“... and g is a probability density function of a random matrix ...”

In the Supplementary Method F, we revised the following sentence:

“... $\text{var}(TG) = (E[T])^2\text{var}(G) + (E[G])^2\text{var}(T) + \text{var}(T)\text{var}(G)$...”

In the “Discussion” section, we revised the following sentence:

“... Recent studies have shown that Genotype-by-Environment Interaction (GEI or $G \times E$) may explain ...”

For the fifth point ($G \sim \text{Binomial}(2; \text{MAF}) \rightarrow G \sim \text{Binomial}(2; \text{AF})$), we think it might be more accurate to keep “MAF” (Minor Allele Frequency) since genotypes in our simulation studies and real data analysis were coded as the number of minor alleles by assuming an additive genetic model.

Response to Reviewer 2

Zhai et al. propose a method to model PRS for PGx studies by simultaneously modeling both the prognostic and predictive effects. They perform simulation studies and show that their newly developed PRS-PGx methods generally outperform the disease PRS methods with PRS-PGx-Bayes being superior. Application of PRS-PGx methods to a large cardiovascular RCT (IMPROVE-IT) to predict treatment related LDL cholesterol showed noticeable improvements.

This study is well-written and provides good data in both simulations and applied to a cardiovascular RCT for improved PRS methods for drug response prediction in PGx studies. These methods are needed, as the authors noted, most PGX PRS studies to date have relied on standard disease PRS methods which relies on stringent assumptions that are not necessarily correct and are restricted in terms of prediction potential and can result in limited power in prediction. By considering a PGx PRS that not only is built from variants with prognostic effects but also predictive effects related to treatment, an optimized PRS can be built to study treatment interaction effects for PRS in PGx studies.

Response: We thank the reviewer for the high-level summary of our research work in this manuscript.

Major comments

1 There has been intense focus on extending disease PRS to multi-ethnic populations. Studies have shown that PRS trained on European GWAS data have limited portability and reduced prediction in Non-European populations. The issue of multi-ethnic populations is absent in this study - how do these PGx PRS studies function with respect to trans-ancestry populations?

Response: Thank you for raising this good point. We agree with the reviewer that in the disease PRS area, there have been quite a few publications/preprints addressing this issue. For example, Márquez-Luna et al. (2017) used data from multiple ethnic groups for training and combined the per ethnic group scores by weighted averaging them into a joint score that can increase the prediction performance for non-Caucasian groups. Ruan et al. (2021) presented another methodology by incorporating a similar weighted average score into the Bayesian regression framework and reported the improvement of the prediction of quantitative traits and schizophrenia risk in non-European populations. Therefore, it is reasonable to assume that a dedicated multi-ethnic PGx-PRS method can improve the drug response prediction in non-Caucasian populations.

To clarify that our methods were developed based on single-ethnic population (e.g., European population), in the “Discussion” section, we added the following sentences:

“... Moreover, our methods are developed based on single-ethnic population (e.g., European population). A direct application of this score to other ethnic groups may result in considerable loss in prediction accuracy. It is, therefore, of great research interest to extend the proposed PRS-PGx approaches to the trans-ethnic scenario in future ...”

References:

- [1] Márquez-Luna, C., Loh, P., South Asian Type 2 Diabetes (SAT2D) Consortium, SIGMA Type 2 Diabetes Consortium, & Price, A. L. Multiethnic polygenic risk scores improve risk prediction in diverse populations. *Genetic Epidemiology* **4**, 811–823 (2017).
- [2] Ruan, Y., Lin, Y., Feng, Y. A. et al. Improving Polygenic Prediction in Ancestrally Diverse Populations. *Preprint at <https://www.medrxiv.org/content/10.1101/2020.12.27.20248738>* (2021).

2 Although I understand this is primarily a methods paper demonstrating better prediction performance of PGx PRS methods for PGx studies, there is hardly any empirical results shown on the PGx PRS-Bayes method applied to the IMPROVE-IT study. Only r^2 and P -values are shown but it would be informative to show effect sizes and association results of PGx PRS-Bayes for the different arms of the IMPROVE-IT study.

Response: Thank you for your suggestion. We agree with you that it is important to add detailed results about the effect sizes. Per your suggestion, we revised Table 1 to include more information (i.e. β s, SEs).

Table 1 IMPROVE-IT PGx GWAS data analysis results: R^2 , p-values, and effect sizes.

PRS Method	Two arms ¹	T arm ²	C arm ³	Two arms	T arm	C arm	Two arms	T arm	C arm
	R^2	R^2	R^2	$Pval_{G \times T}$	$Pval_G$	$Pval_G$	$\hat{\beta}_{G \times T}$ (SE)	$\hat{\beta}_G$ (SE)	$\hat{\beta}_G$ (SE)
PRS-Dis-CT	0.165	0.152	0.191	0.041	3.0e-09	5.1e-17	-0.031 (0.015)	-0.066 (0.011)	-0.035 (0.004)
PRS-Dis-LDpred2	0.174	0.165	0.201	0.033	4.3e-13	6.1e-23	-0.037 (0.017)	-0.079 (0.011)	-0.042 (0.004)
PRS-PGx-Unadj	0.165	0.180	0.121	0.028	1.2e-13	2.7e-03	-0.061 (0.028)	-0.082 (0.011)	0.057 (0.019)
PRS-PGx-CT	0.184	0.241	0.070	0.009	1.7e-15	0.01	-0.095 (0.036)	-0.104 (0.013)	-0.040 (0.016)
PRS-PGx-GL	0.181	0.203	0.123	0.014	7.4e-15	1.8e-03	-0.076 (0.031)	-0.093 (0.012)	0.112 (0.036)
PRS-PGx-Bayes	0.214	0.277	0.194	5.4e-05	3.8e-21	1.0e-17	-0.131 (0.032)	-0.124 (0.013)	0.198 (0.023)

¹ Two-arm model: $Y \sim T + S_{prog} + T \times S_{pred}$

² T-arm model: $Y \sim S_{PGx}$ [= $Y \sim (S_{prog} + S_{pred})$], where $S_{PGx} = S_{prog} + T \times S_{pred}$

³ C-arm model: $Y \sim S_{PGx}$ [= $Y \sim S_{prog}$], where $S_{PGx} = S_{prog} + T \times S_{pred}$

In the “Results: Polygenic prediction of drug responses in the IMPROVE-IT PGx GWAS study” section, we added the following sentences to describe Table 1:

“... Table 1 also shows that the marginal effect sizes $\hat{\beta}_G$ of S_{PGx} from the model $Y \sim S_{PGx}$ under treatment arm were all negative across different PRS-Dis and PRS-PGx methods, indicating that a larger PRS would result in more LDL reduction after 1-month treatment of Ezetimibe + Simvastatin. In the meantime, PRS-PGx-Bayes method outperformed the others with the largest absolute value of effect size $\hat{\beta}_G$. Similarly, the interaction effect sizes $\hat{\beta}_{G \times T}$ of S_{pred} from the model $Y \sim T + S_{prog} + T \times S_{pred}$ were all negative across all methods, implying that a larger predictive score would result in a larger treatment effect (i.e., Ezetimibe + Simvastatin combination vs. Simvastatin monotherapy). PRS-PGx-Bayes method is also superior to the others with the largest absolute value of effect size $\hat{\beta}_{G \times T}$...”

3 Table 1: The five PRS methods have R^2 between 0.15 - 0.277, with the best performing being PRS-PGx-Bayes especially for two arms and T arm, which is what we expect for this method. However, the R^2 values are fairly low across all five PRS methods. Can the authors provide possible reasons for the low R^2 ? Does this indicate that PRS for PGx studies have low prediction in generally currently?

Response: Thank you for your questions. In principle, the prediction R^2 depends on the underlying heritability (i.e., the proportion of phenotypic variability that can be explained by genetics) of the drug response studied. We did not calculate the heritability of Simvastatin/Ezetimibe effect on LDL in this study since it is out of the scope of this manuscript. But our numbers of R^2 (0.15 \sim 0.277) are generally consistent with the heritability of other drug responses or complex traits in

literature. For example, Kalka et al. (2021) studied the heritability of variation in the glycaemic response to metformin, first-line therapeutic agent for type 2 diabetes (T2D), by leveraging 18 years of electronic health records (EHR) data from Israel’s largest healthcare service provider, consisting of over five million patients of diverse ethnicities and socio-economic background. By using Linear Mixed Model-based framework, a common-practice method for heritability estimation, they calculated a heritability measure of $h^2 = 12.6\%$ (95% CI, 6.1% ~ 19.1%) for absolute reduction of HbA1c% after metformin treatment in the entire cohort, $h^2 = 21.0\%$ (95% CI, 7.8% ~ 34.4%) for males and $h^2 = 22.9\%$ (95% CI, 10.0% ~ 35.7%) in females.

In addition, Chhibber et al. (2014) estimated the heritability of cancer drug response by measuring drug cytotoxicity within familial-derived lymphoblastoid cell lines (LCL). They summarized a series of drug response heritability results from cell line experiments in their Table 2 (<https://www.ncbi.nlm.nih.gov/pmc/articles/PMC4308414/>). From that table, we can see the heritability varies from 8.1% to 70% depending on the drug and dose levels.

Even in disease genetic studies, Ge et al. (2019) developed the PRS-CS method and applied it to the Partners HealthCare Biobank data and further compared with other five polygenic prediction methods. Polygenic scores were built to predict the risk of common complex diseases such as breast cancer, coronary artery disease, depression, inflammatory bowel disease, rheumatoid arthritis, and type 2 diabetes mellitus, as well as other quantitative traits like height, body mass index, and lipid levels. They presented the prediction results in their Fig. 2 (<https://www.nature.com/articles/s41467-019-09718-5>). The prediction R^2 are all less than 0.3.

In summary, the prediction R^2 between 0.15 ~ 0.277 from our five PRS-PGx methods should not be considered low, for either drug response or disease, under the polygenic prediction context. This does not indicate that PRS for PGx studies have low prediction in general. Besides the methods that are used to build the PRS, the prediction R^2 really depends on the underlying heritability of the drug response studied.

References:

- [1] Kalka, I.N., Gavrieli, A., Shilo, S. et al. Estimating heritability of glycaemic response to metformin using nationwide electronic health records and population-sized pedigree. *Communications Medicine* **1**, 55 (2021).
- [2] Chhibber, A., Kroetz, D. L., Tantisira, K. G., McGeachie, M., Cheng, C., Plenge, R., Stahl, E., Sadee, W., Ritchie, M. D., & Pendergrass, S. A. Genomic architecture of pharmacological efficacy and adverse events. *Pharmacogenomics* **15**, 2025–2048 (2014).
- [3] Ge, T., Chen, C.-Y., Ni, Y., Feng, Y.-C. A. & Smoller, J. W. Polygenic prediction via Bayesian regression and continuous shrinkage priors. *Nature Communications* **10**, 1776 (2019).

4 Following on comment above, for disease PRS, studies have reported the top X percentile of PRS vs. rest of distribution and highlight X fold odds ratio which can be informative for risk stratification. Can a similar logic be used here?

Response: This is a very good point. Per your suggestion, we revised Fig. 4 in the manuscript as follows.

Fig. 4 Patient stratification performance of six polygenic prediction methods in the IMPROVE-IT PGx real data analysis. **a** Quantile plot of treatment effect using four fixed quantiles (0% ~ 25%, 25% ~ 50%, 50% ~ 75% and 75% ~ 100%). TE stands for “treatment effect”, and CI denotes “confidence interval”. **b** Differential treatment effect when patients were stratified into top 10%, 20%, ..., 90% percentile of the predictive score vs. the rest, respectively.

We also revised the description of Fig. 4 in the “Results: Polygenic prediction of drug responses in the IMPROVE-IT PGx GWAS study” section as follows:

“... We further compared the patient stratification performance across different methods with the results summarized in Fig. 4. In Fig. 4a, we used four fixed quantiles (0% ~ 25%, 25% ~ 50%, 50% ~ 75% and 75% ~ 100%). The results indicated that although overall the population had a positive treatment effect (i.e., Simva+EZ is better), the treatment effects varied across different patient subgroups when stratified by the predictive score. Furthermore, the predictive score determined by PRS-PGx-Bayes was generally superior to other methods for patient stratification. Specifically, ratios of top 75% ~ 100% subgroup to bottom 0% ~ 25% subgroup in terms of treatment effects were 1.27, 1.48, 1.65, 3.27, 1.94, and 10.28 for PRS-Dis-CT, PRS-Dis-LDpred2, PRS-PGx-Unadj, PRS-PGx CT, PRS-PGx-GL, and PRS-PGx-Bayes, respectively. In Fig 4b, patients were stratified into top 10%, 20%, ..., 90% percentile of the predictive score vs. the rest, respectively. The corresponding between group differential treatment effect was calculated. Among the six methods, PRS-PGx-Bayes had the largest differential treatment effect across different cutoff points followed by PRS-PGx-CT and PRS-PGx-GL; and the rest three methods had the lowest differential treatment effects. The optimal cutoff point for PRS-PGx-Bayes occurred between 50% ~ 60%, with differential treatment effect around 0.52. Instead of using fixed quantiles, we also determined the optimal quantile cutoffs with the largest differential treatment effect estimated from the 5-fold cross-validation (training and testing) procedures. The corresponding ability of PRS-PGx-Bayes to stratify patients with greater clinical benefits was assessed in different validation sets with the

results summarized in Supplementary Figure 10. The differences in treatment effects between high and low predictive score subgroups were very clear in the overall population as well as in four out of five CVs ...”

Minor comments

5 *Avoid superfluous words which appear to overhype the message of the study. For example “paradigm shift” in the Abstract could be toned down.*

Response: Thanks for the suggestion. Per your suggestion, we deleted “paradigm” in the Abstract:

“... We propose a ~~paradigm~~ shift from disease PRS to PGx PRS approaches by simultaneously modeling both the prognostic and predictive effects and further make this ~~paradigm~~ shift possible by developing a series of PRS-PGx methods, including a novel Bayesian regression approach (PRS-PGx-Bayes) ...”

REVIEWER COMMENTS

Reviewer #1 (Remarks to the Author):

- I think the authors have carefully answered most of my comments (Reviewer 1), thank you for your efforts. However, I feel like some of the explanations they give should also be mentioned in the manuscript so that it is useful to other readers as well (e.g. for comments #1, #2, #6, and #13).

- The runtimes of the proposed method seem to be really large. Not everyone can run 400 jobs simultaneously. You should make your method more accessible by trying to optimize it a bit.

- "It is, therefore, of great research interest to extend the proposed PRS-PGx approaches to the trans-ethnic scenario in future" -> not just of research interest, but also of public interest..

Reviewer #2 (Remarks to the Author):

The Reviewers have adequately addressed all of my comments. I have no further comments.

We would like to thank two reviewers for carefully reviewing our response letter and giving us additional comments and suggestions. Their input enabled us to significantly improve the quality of our manuscript, which is highly appreciated. Detailed responses to reviewers are given below. We highlighted the changes (in blue) related to the comments in this revision.

Response to Reviewer 1

1 I think the authors have carefully answered most of my comments (Reviewer 1), thank you for your efforts. However, I feel like some of the explanations they give should also be mentioned in the manuscript so that it is useful to other readers as well (e.g. for comments #1, #2, #6, and #13).

Response: Thank you for your comments and suggestions. We are glad to know that the reviewer was satisfied with our responses to most of the comments. We agree with the reviewer that some of the explanations should also be mentioned in the manuscript. Per your suggestion, we mentioned those key explanation points in our responses to the comments #1, #2, #6, and #13 below as well as in the main manuscript.

- Comment #1: explain why we need to adjust for the baseline in the model
 - We revised the following sentences in the “Methods: IMPROVE-IT PGx GWAS data analysis” section:

“... As recommended by Zhang et al. (2022), we adjusted for the baseline LDL-C level Y_0 (in the log scale) in the model to appropriately control the type I error rate (or genome inflation) ...”

- Comment #2: briefly summarize the novelties of the PRS-PGx-Bayes method we proposed in terms of the concept, the model, and the technicality
 - We added the following sentences in the “Discussion” section:

“... To our best knowledge, no existing methods can be directly applied to jointly model both prognostic and predictive effects for drug response prediction. The necessity of using PGx PRS approaches instead of disease PRS approaches is validated by the proof of extremely stringent assumptions needed for the disease PRS approach to predict drug response ...”

“... Compared with the PRS-Dis methods, the PRS-PGx approaches can shrink variants’ main and interaction effect sizes simultaneously, and construct PGx scores including a prognostic PRS and a predictive PRS. Thus, in our PRS-PGx-Bayes method, we propose to accommodate both effects and their correlation by modeling a variance-covariance matrix. Although the inverse Wishart (IW) prior is widely used, in this paper, we choose to use the hierarchical half-t prior (Huang and Wand, 2013) instead due to the limitation of IW (i.e., IW prior imposes a dependency between the correlations and the variances). Moreover ...”

- Comment #6: understand the real data analysis results (i.e., why we got good prediction results with small sample size in IMPROVE-IT PGx GWAS)

– We revised and added the following sentences in the “Discussion” section:

“... On one hand, although sample sizes from PGx studies are usually smaller than those in disease GWAS, the significantly larger effect sizes from PGx studies (Harper and Topol, 2012; Maranville and Cox, 2016) can result in decent power to detect variants associated with drug response phenotypes and further enable good drug response prediction performance of PGx PRS. On the other hand, the availability of summary statistics from PGx GWAS is expected to increase quickly (i.e., similar as the case for the availability of summary statistics from disease GWAS). Therefore, statistical methods customized for PGx PRS analysis are urgently needed for drug response prediction and patient stratification in PGx GWAS studies, with the ultimate goal of achieving precision medicine ...”

- Comment #13: revise the description of PRS-PGx-CT method

– We revised the following sentences in the “Methods: PRS-PGx-CT” section:

“... Specifically, in the clumping step, for any pair of SNPs that have a physical distance smaller than 250 kb and an LD $r^2 > 0.01$, the less significant SNP is removed. Furthermore, in the thresholding step, the prognostic and predictive effect size estimates of SNPs, whose 2-df test (i.e., joint test of $G + G \times T$, obtained from PGx GWAS summary statistics) p-values not passing the threshold P_T , will be shrunk to zero. And then the remaining SNPs are kept with both types of effects ...”

References:

- [1] Zhang, H., Chhibber, A., Shaw, P.M., Mehrotra, D.V., & Shen, J. A statistical perspective on baseline adjustment in pharmacogenomic genome-wide association studies of quantitative change. *npj Genomic Medicine* **7**, 1-10 (2022).
- [2] Harper, A. R., & Topol, E. J. Pharmacogenomics in clinical practice and drug development. *Nature Biotechnology* **30**, 1117–1124 (2012).
- [3] Maranville, J. C., & Cox, N. J. Pharmacogenomic variants have larger effect sizes than genetic variants associated with other dichotomous complex traits. *The Pharmacogenomics Journal* **16**, 388–392 (2016).

The above three references have also been added to the References section.

2 The runtimes of the proposed method seem to be really large. Not everyone can run 400 jobs simultaneously. You should make your method more accessible by trying to optimize it a bit.

Response: Thanks for your comments. Per your suggestion, we carefully inspected the computational performance of our PRS-PGx-Bayes R function and identified two pieces of R code where we

can optimize them for much faster running performance. Specifically, we modified our PRS-PGx-Bayes function in two places.

First, we replaced ‘cov’ and ‘cor’ functions by ‘cova’ and ‘cora’ functions from the R Package **Rfast** (<https://cran.r-project.org/web/packages/Rfast/index.html>), respectively. ‘cova’ and ‘cora’ can perform much faster covariance and correlation matrix calculations compared with the traditional ‘cov’ and ‘cor’ functions. To illustrate that, we conducted additional simulations comparing cor() and cora() on the largest LD block (chr 6, LD block 33). The simulation results were presented in Response Figure 1. After matching IMPROVE-IT PGx data to 1000 Genomes data, we had 11,769 SNPs left in this block. We then evaluated the computational time by randomly choosing 1,000, 3,000, 5,000, 7,000, 9,000 and the full 11,769 SNPs from the block. For the largest LD block, calling cor() took around 3 minutes, while cora() only took around 30 seconds, which is about 6 times faster.

Response Figure 1: Computational time of calling cor() (left panel) and cora() (right panel) on chromosome 6 block 33. Number of variants = 1,000, 3,000, 5,000, 7,000, 9,000, and 11,769 (whole LD block).

Second, we called ‘nearPD’ function to compute the nearest positive definite matrix for an approximate one. And then this matrix was used for Cholesky decomposition to update effect sizes **b**. This procedure was repeated for each iteration (number of iterations = 1000 by default in the PRS-PGx-Bayes function). However, we realized that we only need to call the ‘nearPD’ function when the matrix itself is not positive definite (i.e., ‘is.positive.definite() == FALSE’). ‘is.positive.definite’ is a R function from R Package **matrixcalc**. Adding this judgement condition before downstream analysis significantly reduced the computational time.

With these two modifications, our revised PRS-PGx-Bayes function (named as PRS-PGx-Bayes-v2) is much more efficient. We conducted additional simulations to evaluate the total computational time of PRS-PGx-Bayes vs. PRS-PGx-Bayes-v2 using the largest LD block (i.e., chr 6, LD block 33) with 1,000 MCMC iterations. As shown in Response Figure 2, it took PRS-PGx-Bayes around 43.5 hours for the largest LD block and 3.4 hours for the median LD block; while it took PRS-PGx-Bayes-v2 around 5.9 hours for the largest LD block and 1 hour for the median LD block.

In the authors’ working environment (i.e., the High Performance Computing environment), the IMPROVE-IT whole genome analysis took about 40 hours, where typically 200 - 400 jobs were run simultaneously for PRS-PGx-Bayes function; while it took about 35 hours with 50 jobs running simultaneously for PRS-PGx-Bayes-v2, which is about 10 times faster in total.

Response Figure 2: Computational time on the largest LD block (chr 6, block 33) by running PRS-PGx-Bayes (left panel) and PRS-PGx-Bayes-v2 (right panel) functions with 1,000 MCMC iterations. Number of variants = 1,000, 3,000, 5,000, 7,000, 9,000, and 11,769 (whole LD block). The real genetic data was obtained from the IMPROVE-IT trial with the sample size of 5,661. The effect sizes and phenotype data were simulated with heritability fixed at 0.3, $\psi/\xi = 1$, and $P(\text{causal}) = 0.01$.

To further investigate how these changes in the R code would affect the predictive performance, we re-ran the simulation following the “Methods: Simulations” section in the main text. Simulation settings remained the same as described in Fig. 2. Four criteria were used to compare the performance between PRS-PGx-Bayes and PRS-PGx-Bayes-v2: overall R^2 , R^2 in treatment arm, R^2 in control arm, and predictive p-value. Almost no performance difference was observed under different genetic architectures, which is shown in Response Figure 3.

Response Figure 3: Predictive performance comparisons between PRS-PGx-Bayes and PRS-PGx-Bayes-v2 (with modified R code) in the simulation studies, where heritability was fixed at 0.3 and $\psi/\xi = 1$. The numbers of the causal variants for $P(\text{causal}) = 0.001, 0.01$ and 0.1 were 5, 50 and 500, respectively. The training sample size was 3,000. The error bar represents the standard error of the results calculated from the testing sets. The performance was assessed in the testing set in terms of (a) prediction accuracy R^2 of S_{PGx} in two arms, (b) p-value for the predictive effect of $S_{pred} \times T$ interaction, (c) R^2 of S_{PGx} under treatment arm, and (d) R^2 of S_{PGx} under control arm.

Finally, we revised the following sentences in the “Results: Simulation studies: Computational time” section:

“... The result also shows that it took roughly 5.9 hours for the largest LD block and 1 hour for the median-size LD block (Supplementary Figure 10) to complete the computation ... In the authors’ High Performance Computing working environment, the IMPROVE-IT whole genome analysis took about 35 hours, where typically 50 jobs were run simultaneously ...”

We also updated the Supplementary Figure 9 accordingly:

Supplementary Figure 9: Computational time on the largest LD block (chr 6, block 33) by running PRS-PGx-Bayes function with 1,000 MCMC iterations. Number of variants = 1,000, 3,000, 5,000, 7,000, 9,000, and 11,769 (whole LD block). The real genetic data was obtained from the IMPROVE-IT trial with the sample size of 5,661. The effect sizes and phenotype data were simulated with heritability fixed at 0.3, $\psi/\xi = 1$, and $P(\text{causal}) = 0.01$.

We will update the RS-PGx-Bayes function as mentioned above in the next version of our PRSPGx R package on CRAN. For the purpose of this review, we also provide the R package source file PRSPGx_0.2.0.tar.gz file for your reference in this round of submission.

3 “It is, therefore, of great research interest to extend the proposed PRS-PGx approaches to the trans-ethnic scenario in future” → not just of research interest, but also of public interest.

Response: Thanks for pointing this out. Indeed, as the efforts to diversify samples in genomic studies start to grow, the scale of non-European genomic resources has been expanded in recent years. Therefore, there is an urgent need to improve the accuracy of cross-population polygenic prediction in order to maximize the clinical potential of PRS. Per your suggestion, we revised the following sentences in the “Discussion” section:

“... With the rapid growth of non-European genomic resources in recent years, it is, therefore, of great both research and public interests to extend the proposed PRS-PGx approaches to the trans-ethnic scenario in future ...”

Response to Reviewer 2

The authors have adequately addressed all of my comments. I have no further comments.

Response: Thank you for your previous comments and questions again. We are glad that the reviewer was satisfied with our responses.

REVIEWERS' COMMENTS

Reviewer #1 (Remarks to the Author):

The authors have adequately addressed all of my comments. I have no further comments.